# Towards 3D Objectness Learning in an Open World

**Taichi Liu[1], Zhenyu Wang[2], Ruofeng Liu[3], Guang Wang[4], Desheng Zhang[1]**
[1]Rutgers University [2]Tsinghua University [3]Michigan State University [4]Florida State University
`taichi.liu@rutgers.edu, wangzy20@tsinghua.edu.cn`
`liuruofe@msu.edu, guang.wang@fsu.edu, desheng@cs.rutgers.edu`
op3det.github.io

## Abstract

Recent advancements in 3D object detection and novel category detection have made significant progress, yet research on learning generalized 3D objectness remains insufficient. In this paper, we delve into learning open-world 3D objectness, which focuses on detecting *all* objects in a 3D scene, including novel objects unseen during training. Traditional closed-set 3D detectors struggle to generalize to open-world scenarios, while directly incorporating 3D open-vocabulary models for open-world ability struggles with vocabulary expansion and semantic overlap. To achieve generalized 3D object discovery, we propose **OP3Det**, a class-agnostic **O**pen-World **P**rompt-free **3D Det**ector to detect any objects within 3D scenes without relying on hand-crafted text prompts. We introduce the strong generalization and zero-shot capabilities of 2D foundation models, utilizing both 2D semantic priors and 3D geometric priors for class-agnostic proposals to broaden 3D object discovery. Then, by integrating complementary information from point cloud and RGB image in the cross-modal mixture of experts, OP3Det dynamically routes uni-modal and multi-modal features to learn generalized 3D objectness. Extensive experiments demonstrate the extraordinary performance of OP3Det, which significantly surpasses existing open-world 3D detectors by up to 16.0% in AR and achieves a 13.5% improvement compared to closed-world 3D detectors.

## 1 Introduction

In 3D perception systems, especially in real-world environments such as autonomous driving and robotics, object categories of interest may change dynamically. This has led to increasing attention on challenging tasks like out-of-distribution 3D detection [1, 2], open-world 3D detection [3, 4] and open-vocabulary 3D detection [5, 6, 7, 8, 9, 10], improving generalization beyond closed-set assumptions. A core challenge across these tasks is the ability to localize *all* objects, which lies in understanding how objects are structured in 3D scenes to distinguish them from the background. While extensive efforts have been made to identify unknown or novel objects in the 2D domain [11, 12, 13] - known as class-agnostic object detection (OD) [14], such exploration in the 3D domain remains limited. To bridge this gap, we introduce learning 3D objectness in a class-agnostic paradigm, enabling models to detect and discover objects beyond known categories. Therefore, our goal is to achieve *class-agnostic 3D object detection*, where objects are identified and localized based on their intrinsic properties rather than pre-defined semantic labels, thus supporting open-world perception.

In a class-agnostic manner, ensuring a high recall rate is essential, as it ensures that the majority of objects in the scene are detected regardless of their semantic categories. This serves as a foundation for accurate category assignment and significantly contributes to object detection for categories of interest [7, 15]. Although current point-cloud-based 3D detectors [16, 17, 18, 19, 20] have achieved significant success in 3D benchmark datasets [21, 22, 23, 24], simply shifting from class-specific to class-agnostic classification is ineffective. This is primarily because 3D point cloud

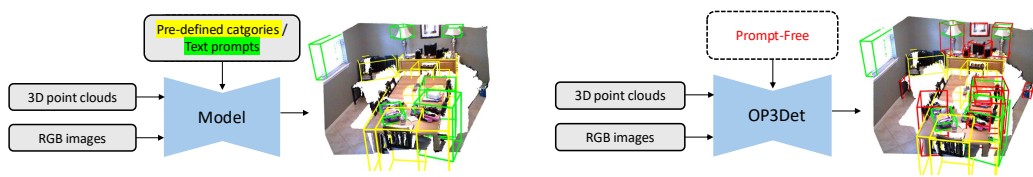

(a) existing 3D object detectors                    (b) OP3Det (ours)

Figure 1: **Illustration for Prompt-free 3D objectness learning.** (a) Closed-world detectors can only recognize pre-defined categories in the training dataset (yellow boxes). Although some 3D detectors can detect novel classes via pre-defined prompts (green boxes), they still cannot detect "all" when the given vocabularies are limited. (b) In comparison, our OP3Det can detect rare categories (red boxes) and better discover 3D objects without the requirement of any semantic labels and text prompts.

data are extremely limited in both the scale of data and annotated categories. Moreover, directly employing open-vocabulary 3D models for class-agnostic detection faces significant challenges due to vocabulary expansion and semantic overlap [25] in hand-crafted text prompts, making them ineffective for novel object discovery and preventing the learning of open-world 3D objectness, as can be seen in Fig 1. Therefore, learning 3D open-world objectness and achieving strong localization generalization is highly challenging. In contrast, the 2D domain is far more resource-rich in both models and data. Plenty of pre-trained foundation models [26, 27] and the detectors trained on extensive vocabularies [28, 29, 30] with broad classes [31, 32, 33] demonstrate strong generalization capability. Our intuition is to transfer strong zero-shot abilities from 2D pre-trained models to the 3D domain, exploring its generalization ability for 3D object discovery and 3D objectness learning.

We propose **OP3Det**, a class-agnostic **O**pen-World **P**rompt-free **3D Det**ector, which exploits extensive 2D semantic knowledge to learn open-world 3D objectness. Here, prompt-free means that our method requires no text prompts or any semantic priors as inputs at inference time, making it semantic prompt-free—the model directly learns 3D objectness from geometric and visual cues. More specifically, we primarily use the large 2D foundation model - Segment Anything Model (SAM) [27], to extract abundant and generalizable class-agnostic object masks in a scene. However, SAM often produces fragmented masks or partial object masks, which will severely hinder the learning of whole objectness [34, 35]. To address this, we adopt a multi-scale point sampling strategy that considers 3D spatial proximities to refine the uniformly distributed point prompts provided to SAM, enabling more accurate extraction of class-agnostic object bounding boxes. Through semantic and geometric cues, a greater variety of novel objects can be discovered effectively, which are subsequently projected into the 3D space for 3D object discovery in point clouds prior to training.

To better learn 3D objectness during the training phase, we further leverage 2D semantic knowledge and integrate both point cloud and RGB image modalities for multi-modal training. Prior works have explored fusion at various levels, including point-level [36, 37], feature-level [38, 39, 40], and object-level [41, 42, 43, 44]. Although these methods have shown strong performance, they often rely heavily on fused features, while overlooking the importance of preserving modality-specific informative cues. We thus propose the *cross-modal mixture of experts (MoE)* to effectively connect both intra-modal and inter-modal information. Specifically, we use the self-attention structure to encode uni-modal and multi-modal features. Through a multi-modal router and modal-specific experts, OP3Det dynamically fuses uni-modal or multi-modal features, ensuring that the most relevant information can be adopted. The model can adapt its strategy according to the specific demands of each scenario, whether it requires a stronger reliance on 2D semantic information from images, 3D geometric cues from point clouds, or a balanced integration of both modalities.

Our main contributions can be summarized as follows:

- We introduce a novel and practical problem setting, class-agnostic open-world 3D object detection, which aims to detect all objects in a 3D scene and reflect real open-world environments. To the best of our knowledge, we are the first to formally define and address this problem in the 3D domain.
- We propose OP3Det, a multi-modal 3D detector for learning open-world 3D objectness. A multi-scale point sampling strategy is designed to enhance 2D-3D association and reveal a broader range of object instances for effective open-world 3D object discovery.

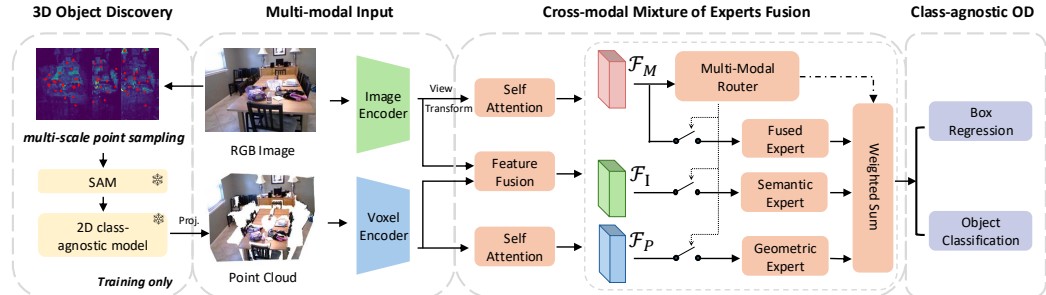

Figure 2: **The overview of OP3Det.** We apply SAM to introduce abundant 2D semantic knowledge for 3D object discovery. Multi-scale point sampling is utilized in this process. The cross-modal MoE is then employed to guide data pathways for uni-modal and multi-modal features, allowing the model to dynamically adapt its reliance on unimodal or cross-modal information according to the scenario.

- We design a cross-modal mixture-of-experts (MoE) module guided by a multi-modal router, which dynamically selects between uni-modal and multi-modal pathways to adaptively learn 3D objectness under diverse open-world scenarios.

Extensive experiments demonstrate the ability of our OP3Det to detect in the open world. OP3Det possesses a strong generalization ability in both cross-category and cross-dataset settings. It achieves 27% improvement for novel class discovery compared to the baseline method. The adaptability of our method also makes it easily extendable to outdoor scenes, class-specific detection or the 2D domain.

## 2  Related Work

**Open-world 3D scene understanding.** Open-world 3D learning aims to identify and detect 3D objects from an arbitrary set of categories, allowing models to generalize to novel object categories that are not present in the training data. Recent methods [45, 46, 47, 48, 49, 50] have conducted open-world learning for 3D segmentation. However, these methods rely on precise mask-level annotations for geometric information learning. Open-vocabulary 3D detectors [5, 6, 7, 8, 9, 10] usually use RGB images and pre-trained 2D models [26, 27] to enrich semantic information for recognizing novel categories. Despite their success, these methods rely on a pre-defined vocabulary as input for detection, rather than truly learning objectness. When the vocabulary is incomplete or mismatched with the scene, they still fail to detect all objects, resulting in a low recall in novel classes. In comparison, we formally explore open-world 3D objectness learning in a class-agnostic way, aiming to detect all salient objects in a scene without relying on a fixed label set or text prompts.

**Applications of SAM in 3D scenes.** The strong zero-shot generalization capabilities of SAM have motivated its adoption in 3D scenes. Previous works leverage SAM to generate fine-grained 3D masks for 3D segmentation. SAM3D [46] and Segment3D [51] both used a bottom-up framework that applied SAM to RGB-D images to obtain 2D masks, which are then projected into 3D space for supervised training. In contrast, methods such as REAL [52], SAM-Graph [53] and OpenMask3D [54] adopt a top-down strategy. They utilize projected 3D labels as prompts to guide SAM in generating more accurate 2D masks, which are subsequently back-projected to produce dense or diverse 3D annotations. However, the segments from SAM are not solely focused on objects. Directly using SAM will introduce noise into the generated 3D labels. Our method apply SAM for 3D object detection in open-world learning. By eliminating the need for 2D text and 3D labels as prompts, we enable scalable training and generalization to unseen objects in an open world.

**2D and 3D Feature Fusion.** Existing multi-modal fusion methods can be generally divided into point-level, feature-level and object-level categories. Point-level fusion introduces 2D features directly in the 3D domain. PointPainting [36] and PointAugmenting [37] enhance LiDAR-based 3D object detection by enriching point cloud features with image semantics. Feature-level fusion integrates multi-modal features using shared representation spaces or attention-based modules, like BEVFusion [38, 39] projecting LiDAR and image features into the BEV space. Object-level fusion integrates modality-specific information at the instance level. SparseFusion [41] fuses instance-level sparse features from both 2D and 3D inputs, and ObjectFusion [42] uses a heatmap-based proposal

generator to align object-centric features. However, these methods often rely heavily on fused features, while overlooking the need to preserve modality-specific critical cues. OP3Det adaptively filters irrelevant cross-modal features while preserving and enhancing informative intra-modal signals.

## 3 Method

We formulate the class-agnostic open-world 3D object detection task in Sec. 3.1. Fig. 2 shows the overall architecture of the proposed OP3Det. To achieve open-world and class-agnostic 3D detection, OP3Det learns 3D objectness through two key components: (i) a 3D Object Discovery strategy (Sec. 3.2) that expands the set of potential 3D objects and (ii) a cross-modal MoE module (Sec. 3.3) that dynamically fuses semantic and geometric representations for robust 3D objectness learning.

### 3.1 Problem Formulation

In our work, 3D objectness denotes the likelihood that a spatial region corresponds to a physically discrete object, distinguishable from background structures or noise, regardless of semantic category. Formally, let $\phi(\cdot)$ denote a learnable model that maps input features $F_{\text{input}}$ to an objectness confidence score. The 3D objectness learning can be expressed as: $I\left[\phi(F_{input}) > \tau\right]$ where $\tau$ is a confidence threshold for classifying a spatial region as foreground, and $I[\cdot]$ is the indicator function (1 denotes a valid object region and 0 denotes background). The model $\phi$ is trained to approximate this decision function, assigning high confidence to true object regions and low confidence to background or noise.

Given a point cloud $X_P$ and corresponding RGB images $X_I$, the training data contain annotated 3D bounding boxes $\{(c_i, bb_i^{3D})\}_{i=1}^M$, where $c_i$ and $bb_i^{3D}$ are the objectness label and 3D bounding box of the i-th object, M is the number of 3D boxes. Our goal is to leverage the paired multi-modal input $(X_P, X_I)$ as input features $F_{\text{input}}$—together with the $bb_i^{3D}$—to learn a detector capable of discovering and localizing all object instances during inference, including novel and unseen categories.

For multi-modal training, the 3D point cloud features $F_P \in \mathbb{R}^{C \times X \times Y \times Z}$ and 2D image features $F_I \in \mathbb{R}^{C \times H \times W}$ are extracted through the voxel-based backbone and the image backbone separately. $F_I$ is then projected into the 3D voxel space for the image features in the voxel space $F_I' \in \mathbb{R}^{C \times X \times Y \times Z}$. Denote the camera intrinsic matrix as $K$ and the extrinsic matrix as $R_t$, then the corresponding positions in the 2D image can be obtained by projecting 3D positions in the 3D voxel space through $KR_t$. We concatenate these two features to obtain the multi-modal features: $F_M = [F_P, F_I']$. For multi-modal fusion, we propose the cross-modal MoE to fuse $F_P, F_I', F_M$, in order to integrate 2D semantic, 3D geometric, and multi-modal information in the training phase.

### 3.2 3D Object Discovery

3D object discovery enables the discovery of novel objects prior to training. To achieve this, we leverage cross-priors from both the 2D and 3D space. In terms of the 2D domain, we utilize the Segment Anything Model (SAM) [27], which is trained on extensive 2D datasets and thus demonstrates strong zero-shot generalization performance across various scenarios. We apply SAM directly on RGB images $X_I$ to conduct segment-anything inference, obtaining a series of class-agnostic masks. Due to the rich semantic information from SAM, these masks often cover a broader range of objects, thus significantly addressing the limitations of object information in 3D datasets. The segment-anything inference process employs a regular 64x64 grid of points $\{(x, y)\}$ as non-semantic point prompts, which serve as inputs to SAM to obtain segmentation results.

However, SAM, as a segmentation model, often leads to fragmented outputs or object parts and sub-parts in the final mask outputs. As illustrated in Fig. 3a, this produces a large number of chaotic masks, introducing significant noise into the final annotations. This severely impedes class-agnostic detection, which generally targets entire objects at a global level. To address this issue, we propose a multi-scale sampling strategy, guided by per-point object prior probabilities. We begin by selecting a source point $(x_s, y_s)$ from the point set that is most likely to be related to the object, according to the IoU score from SAM and attention values from self-supervised model as 2D object priors. Then we filter neighboring points whose 3D distances to the selected point are within a threshold $\delta$, ensuring that local geometric consistency is preserved—a property that cannot be reliably derived from 2D image alone. Specifically, to obtain the 3D distance between the point $(x, y)$ and the source point $(x_s, y_s)$, we project all 3D points onto the 2D image plane through $KR_t$. Then, we select

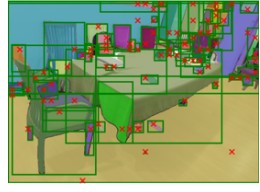 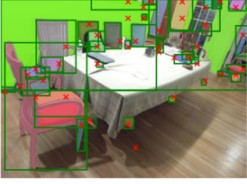 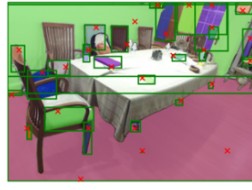 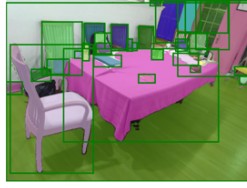

| (a) SAM result | (b) Point sampling, $\tau = 0.2$ | (c) Point sampling, $\tau = 5$ | (d) Multi-scale sampling + class-agnostic detector |

Figure 3: **Visualization of point sampling trategy.** The segmentation masks from SAM contain many small fragments and object parts. By using multi-scale point sampling, these noisy masks can be mitigated. Post-processing with a 2D class-agnostic detector further improves the quality.

the 3D points $(x', y', z')$ and $(x'_s, y'_s, z'_s)$ whose projected points are closest to $(x, y)$ and $(x_s, y_s)$ respectively. The 3D distance between $(x, y)$ and $(x_s, y_s)$ is actually the 3D distance between $(x', y', z')$ and $(x'_s, y'_s, z'_s)$. The iteration continues until no further points are selected. In this way, points that are too close to each other and with low object prior values are filtered out, enabling the elimination of many overly small object masks or object parts, thus reducing noise, which can be seen in Fig. 3b and Fig. 3c.

The choice of $\delta$ is a crucial parameter, as it affects the scale of the final masks. When $\delta$ is too small, filtering of object parts may be insufficient, while a large $\delta$ may lead to the exclusion of useful objects. To address this, we use a series of $\delta$ values ranging from small to large, (0.2,0.5,1,2) in our experiments specifically, and combine their results through NMS. Such a multi-scale point sampling strategy captures the advantages of different scales, yielding more reliable segmentation results.

Finally, to further filter out the remaining small noise masks, such as tiny fragments left when $\delta$ is small, and to enhance object localization, we pass the segmentation results through a pre-trained class-agnostic 2D detector [55]. Since this 2D detector is trained in a class-agnostic way, it focuses on the localization information and is sensitive to complete object boundaries. Thus, it can effectively help determine whether each mask represents a whole object. The object masks and bounding boxes will also be adjusted according to the bounding box regression of such a 2D detector. For objectness prediction, we multiply the IoU prediction scores from SAM and the objectness scores from the class-agnostic 2D detector to obtain the updated scores. We then filter low-score object masks based on such updated scores to ultimately reduce noisy masks, as is illustrated in Fig. 3d. These 2D boxes are ultimately projected into the 3D space through $KR_t$ for 3D object discovery. [1]

### 3.3 Cross-Modal MoE

In the previous subsection, we primarily focus on 3D object discovery in point clouds prior to training. Further, during the training process, we continue to exploit semantic knowledge, integrating geometric information from 3D point clouds to facilitate 3D objectness learning. Therefore, we employ the multi-modal training approach, using both point clouds and RGB images for 3D object detection.

In the closed-world setting, directly using multi-modal features $F_M$ for detection can already lead to performance gains [56, 38]. However, this does not work in the open-world setting. This is because in the class-agnostic binary classification mode, recognizing different objects also heavily relies on geometric information, which is widely present in point cloud features $F_P$. Furthermore, certain multi-modal scenes may be dominated by a single modality, leading to incomplete spatial understanding under occlusions or restricted viewpoints, making it essential to incorporate effective intra-modal interactions to fully exploit the strengths of each individual modality. To address this, we propose a cross-modal Mixture-of-Experts (MoE) module that selectively guide the data pathways of 2D semantic features, 3D geometric features, and multi-modal fused features, achieving dynamic multi-modal fusion to boost 3D objectness learning.

We first utilize the self-attention [57] module on uni-modal and multi-modal features, respectively, on its spatial dimensions. This enables the detector to concentrate on the important spatial regions in

---

[1]The operation is that we project 3D points into the 2D space using $KR_t$, finding points within the 2D box, then clustering them to obtain the 3D box $(\{\hat{bb}_i^{3D}\}_{i=1}^N)$ . This can be viewed as 2D→3D projection.

the features for the subsequent 3D detection, thus extracting important features for each modality: $\mathcal{F}_P = \text{SelfAttn}(F_P)$, and $\mathcal{F}_I$, $\mathcal{F}_M$ are defined and obtained in the same way.

Then, we utilize the multi-modal router to obtain the routing probability $p_P, p_I, p_M$ for different modality features, guiding the data pathways for each modality. This router consists of a 3D convolution layer, a global average pooling layer, a fully connected layer, and the final softmax. Denote the router as $\mathcal{R}$, this process can be denoted as:

$$(p_P, p_I, p_M) = \mathcal{R}(\mathcal{F}_M) \tag{1}$$

Guided by these routing probabilities, we finally apply a semantic expert $\mathcal{E}_I$, geometric expert $\mathcal{E}_P$, and a fused expert $\mathcal{E}_M$. We implement experts through three 3D convolution layers with kernel sizes of 1, 3 and 1 sequentially. The specific process is as follows:

$$\mathcal{F} = \sum_{i \in (P,I,M)} p_i \cdot \mathcal{E}_i(\mathcal{F}_i) \tag{2}$$

Finally, $\mathcal{F}$ is fed into the detection head, where we adopt the 3D detection transformer [58, 15]. As $\mathcal{F}$ represents a synthesis of $F_P$, $F_I$ and $F_M$, the model can dynamically adjust the data pathways based on the input data for dynamic multi-modal fusion, ensuring that the most suitable features can be utilized for the final detection. This thus enhances multi-modal class-agnostic detection.

### 3.4 Training and Inference

**Training.** We use RGB images and point clouds pairs to guide the training of our class-agnostic 3D network. For each image, 3D object discovery is performed using the SAM [27], selected for its large-scale open-world training, remarkable zero-shot generalization to unseen objects, and class-agnostic design. With both annotated and discovered 3D bounding boxes enriched by corresponding RGB images, we then employ the Cross-Modal MoE to train a multimodal 3D detector capable of learning class-agnostic objectness across modalities. Ultimately, the learning loss function of OP3Det primarily follows the loss function in [9]. To better suit our task, the classification loss is formulated as a class-agnostic binary classification loss. For 3D scenes with multi-view images, we extract features from each view and project them into the voxel space using their respective projection matrices. The projected features are then aggregated to the multi-modal representation.

**Inference.** During training, our model utilizes 2D images to discover potential objects and provide semantic supervision for 3D objectness learning. During inference, it performs detection directly on point cloud–image pairs, requiring no additional stages or external modules beyond a standard multi-modal 3D detector. The learned cross-modal MoE further enables class-agnostic 3D objectness inference in a fully prompt-free manner.

## 4 Experiments

**Datasets.** For indoor scenes, we utilize SUN RGB-D [21] and ScanNet V2 [22] datasets. SUN RGB-D contains 46 classes, while ScanNet V2 contains 200 categories in total [59]. We mainly follow the setting of [7] for category splitting. Specifically, for SUN RGB-D, the categories with the top 10 most training samples are selected as base (seen) categories, while the remaining 36 are novel classes. For ScanNet, we also adopt the same setting, using single-view small scenes corresponding to individual images for training. The top 10 classes are utilized for base classes and the other 50 ones for novel classes. Their category labels are removed during training for class-agnostic classification. For outdoor 3D detection, we mainly conduct experiments on the KITTI [23] dataset. We treat the car class as the base class and the cyclist and pedestrian classes as novel classes. We mainly utilize its official metric, the $AP_{70}$ metric with 40 recall positions for evaluation.

Since the target is to identify all objects within a scene for 3D objectness learning and 3D object discovery, and not all bounding boxes are necessarily annotated in the test set, we mainly employ **Average Recall (AR)** under IoU thresholds of 0.25. Average precision (AP) is also utilized. However, under class-agnostic binary classification, AP for base and novel classes cannot be straightforwardly computed, so we only report AP across all categories. For more discussions and experimental results about the AP metric, please refer to the Appendix C.

Table 1: **The cross-category performance of OP3Det on the SUN RGB-D and ScanNet dataset.** Closed-world 3D detection methods are trained on 3D point clouds with only seen categories annotated. Open-vocabulary methods are trained on 3D point clouds with class-specific 3D bounding boxes for annotations, thus requiring more information compared to our class-agnostic setting.

| Method | SUN RGB-D | | | | ScanNet | | | |
|---|---|---|---|---|---|---|---|---|
| | $AR_{novel}$ | $AR_{all}$ | $AR_{base}$ | $AP_{all}$ | $AR_{novel}$ | $AR_{all}$ | $AR_{base}$ | $AP_{all}$ |
| *closed-world 3D object detection methods* | | | | | | | | |
| VoteNet [16] | 33.7 | 68.3 | 79.1 | 55.1 | 35.3 | 44.6 | 56.1 | 13.8 |
| GroupFree [60] | 41.8 | 69.9 | 78.7 | 49.2 | 32.1 | 40.9 | 51.8 | 9.4 |
| FCAF3D [17] | 65.3 | 86.5 | 92.7 | 62.0 | 61.7 | 71.3 | 83.2 | 24.7 |
| Uni3DETR [15] | 51.8 | 82.1 | 91.6 | 61.3 | 54.6 | 67.6 | 80.1 | 16.9 |
| Tr3D [56] | 62.1 | 84.8 | 91.9 | 53.4 | 47.1 | 58.1 | 71.6 | 17.2 |
| *open-vocabulary 3D object detection methods* | | | | | | | | |
| Det-PointCLIPv2 [8] | 22.4 | 31.1 | 64.5 | 10.2 | 33.1 | 38.7 | 55.9 | 3.1 |
| 3D-CLIP [26] | 23.6 | 32.3 | 66.8 | 25.7 | 32.9 | 36.2 | 55.5 | 5.6 |
| CoDA [7] | 33.9 | 60.2 | 71.5 | 48.2 | 44.3 | 53.4 | 68.3 | 23.9 |
| OV-Uni3DETR [9] | 62.8 | 82.5 | 88.8 | 57.4 | 67.6 | 71.6 | 76.5 | 25.9 |
| ImOV3D [61] | 46.9 | 63.1 | 74.1 | 28.3 | 56.9 | 70.6 | 77.9 | 25.0 |
| *class-agnostic open-world 3D object detection method* | | | | | | | | |
| **OP3Det (ours)** | **78.8** | **89.7** | **93.1** | **65.4** | **79.9** | **83.2** | **87.3** | **28.6** |

**Implementation Details.** We implement with mmdetection3D [62], and train with the AdamW [63] optimizer. We use ResNet50 [64] and FPN [65] for the image feature extractor, and sparse 3D ResNet for the voxel feature extractor. We use the multi-scale of $\delta = (0.2, 0.5, 1, 2)$. $N_{point}$ is set to the half number of the total points. We utilize the 0.6 threshold to filter low-quality discovered 3D objects.

**Baselines.** Since class-agnostic 3D detection in the open world has not yet been explored, we compare OP3Det with methods from the related fields of closed-world 3D detection (*i.e.* traditional fully-supervised 3D detection) [66, 60, 17, 15, 56] and open-vocabulary 3D detection [7, 9], and adapt baselines accordingly. For closed-world 3D detection methods, we convert all seen categories into a single class label during training so that these SOTA supervised methods only classify the bounding boxes are objects are not. For open-vocabulary methods, we similarly construct a class-agnostic training setting by replacing all class-specific text prompts with "object". This ensures the model learns to detect general objects without relying on specific category semantics.

## 4.1 Cross-Category Generalization

As is shown in Tab. 1, OP3Det demonstrates significant improvements over existing methods. For novel class discovery, $AR_{novel}$ increases by 13.5% compared to the state-of-the-art closed-world 3D detector FCAF3D, and by 16% compared to the open-vocabulary 3D detector OV-Uni3DETR. Furthermore, our model also shows improvements on base classes, without any decline. This thus contributes to an overall increase in AR across all classes. It is worth mentioning that since novel class objects only represent a small proportion of the scenes, their impact on the overall AR and AP is relatively limited. Nevertheless, even under these conditions, our method still achieves an average increase of over 3% in $AR_{all}$ and $AP_{all}$, strongly demonstrating the 3D object discovery capability of our method. Such substantial improvement highlights that 3D objectness learning addresses a challenging issue about the ambiguous nature of class-agnostic 3D object detection tasks that remains unsolved by existing 3D models, underlining the importance and necessity of our new intuition.

On the larger-scale ScanNet dataset, where the number of categories is higher, our OP3Det continues to demonstrate strong performance, with a clear advantage over existing methods, achieving a 12.3% improvement in $AR_{novel}$. This further validates the capability of our model in the large-vocabulary setting. Notably, under these conditions, the performance gap between base and novel classes is even smaller, highlighting the strong cross-category generalization ability of OP3Det. Compared to the traditional closed-world 3D detectors, our method benefits from leveraging 2D semantic knowledge for 3D object discovery, effectively mitigating the limitations of category information in 3D point clouds. In contrast to open-vocabulary 3D detectors, our use of class-agnostic classification aligns more closely with the objectives of 3D objectness learning. Additionally, the cross-modal MoE effectively integrates multi-modal information, allowing the most relevant features to be applied for class-agnostic detection. The significance of the class-agnostic open-world 3D detection problem as a valuable new direction can also be validated.

Table 2: **The cross-dataset performance of OP3Det on the SUN RGB-D and ScanNet dataset for class-agnostic open-world 3D object detection.** We directly test the trained model on another dataset to obtain the below cross-dataset results.

| Method | ScanNet → SUN RGB-D | | | | SUN RGB-D → ScanNet | | | |
|---|---|---|---|---|---|---|---|---|
| | $AR_{25}$ | $AR_{50}$ | $AP_{25}$ | $AP_{50}$ | $AR_{25}$ | $AR_{50}$ | $AP_{25}$ | $AP_{50}$ |
| *closed-world 3D object detection methods* | | | | | | | | |
| VoteNet [16] | 34.8 | 2.0 | 10.8 | 0.1 | 30.4 | 6.3 | 9.6 | 1.2 |
| GroupFree3D [60] | 41.4 | 0.4 | 1.9 | 0.1 | 39.4 | 5.2 | 8.7 | 0.1 |
| FCAF3D [17] | 59.3 | 8.1 | 17.9 | 0.6 | 47.7 | 14.6 | 12.9 | 1.9 |
| Uni3DETR [15] | 51.3 | 6.4 | 11.9 | 0.2 | 45.7 | 10.9 | 11.3 | 1.3 |
| Tr3D [56] | 54.6 | 4.5 | 11.4 | 0.2 | 45.2 | 10.7 | 9.4 | 1.6 |
| *open-vocabulary 3D object detection methods* | | | | | | | | |
| CoDA [7] | 21.4 | 2.8 | 6.2 | 0.1 | 32.7 | 5.2 | 8.9 | 0.4 |
| OV-Uni3DETR [9] | 49.5 | 3.2 | 8.1 | 0.3 | 52.0 | 15.4 | 9.5 | 0.8 |
| *class-agnostic open-world 3D object detection method* | | | | | | | | |
| **OP3Det (ours)** | **73.1** | **10.7** | **22.3** | **1.1** | **77.9** | **37.3** | **21.2** | **5.1** |

Table 3: **The performance of OP3Det on KITTI dataset for class-agnostic open-world 3D object detection.** The models are trained only on car and are evaluated on car, pedestrian and cyclist. We report $AP_{70}$ with 40 recall positions. *: $AP_{3D}$ on the moderate difficulty is the most important metric.

| Method | $AP_{3D}$ | | | $AP_{BEV}$ | | |
|---|---|---|---|---|---|---|
| | easy | medium* | hard | easy | medium | hard |
| *closed-world 3D object detection methods* | | | | | | |
| SECOND [69] | 61.05 | 62.36 | 61.36 | 63.15 | 69.00 | 68.46 |
| PointPillar [70] | 59.54 | 62.13 | 60.04 | 63.04 | 68.87 | 66.75 |
| Part-$A^2$ [71] | 61.28 | 63.43 | 63.57 | 62.93 | 69.04 | 69.88 |
| 3DSDD [72] | 61.42 | 62.34 | 62.06 | 62.93 | 68.50 | 68.29 |
| PV-RCNN [73] | 59.88 | 65.18 | 65.67 | 63.01 | 69.36 | 70.42 |
| Uni3DETR [15] | 63.54 | 65.74 | 65.43 | 62.74 | 69.01 | 69.87 |
| *open-vocabulary 3D object detection method* | | | | | | |
| OV-Uni3DETR [9] | 62.66 | 63.20 | 62.82 | 64.33 | 69.15 | 68.98 |
| *class-agnostic open-world 3D object detection method* | | | | | | |
| **OP3Det (ours)** | **63.56** | **66.75** | **66.42** | **65.13** | **71.37** | **70.34** |

## 4.2 Cross-Dataset Generalization

Moreover, since class-agnostic open-world 3D detection aims for robust performance in unseen or unknown domains, we further validate the cross-domain detection capability through cross-dataset experiments. In this case, point clouds in different datasets are generally collected through varying methods or sensors. Specifically, SUN RGB-D provides point clouds directly captured by single-view RGB-D cameras, whereas ScanNet reconstructs point clouds from multi-view RGB-D image sequences. As a result, the two datasets exhibit substantial differences in the structure and content of their point clouds, making cross-dataset evaluation in the 3D domain considerably more challenging.

To this end, we conduct cross-dataset experiments for both the SUN RGB-D → ScanNet and ScanNet → SUN RGB-D settings. The results are presented in Tab. 2. Due to differences in category definitions among datasets [67, 68], we only measure AR and AP across all objects. As can be seen, for many existing 3D detectors, a significant performance drop appears in cross-dataset scenarios. For instance, CoDA performance deteriorates noticeably. This is largely due to the reliance of some 3D detectors on point-based backbones for feature extraction, making them highly dependent on dataset-specific geometric information, which limits their effectiveness in cross-dataset generalization. Closed-world 3D detectors suffer from limited supervision due to restricted annotations, while open-vocabulary 3D detectors with class-specific classification are vulnerable to category definition conflicts. In contrast, our method demonstrates substantial performance gains, achieving 30% $AR_{25}$ improvement in the SUN RGB-D → ScanNet setting. The $AP_{25}$ improvement is also almost 10%. Besides, the cross-dataset performance also closely approaches in-dataset results, with only 2% lower $AR_{25}$. This further confirms the strong cross-dataset generalization ability of our method and its effectiveness in learning open-world 3D objectness. Through both cross-category and cross-dataset evaluation, the strong 3D object detection capability of OP3Det in indoor scenes can be demonstrated.

Table 4: **Comparison with 3D open-vocabulary methods on the SUN RGB-D and ScanNet dataset for open-vocabulary 3D object detection (class-specific).** The experimental setting is totally the same as CoDA, and the utilized data are downloaded from CoDA officially released code.

| Method | SUN RGB-D | | | ScanNet | | |
|---|---|---|---|---|---|---|
| | $AP_{novel}$ | $AP_{base}$ | $AP_{all}$ | $AP_{novel}$ | $AP_{base}$ | $AP_{all}$ |
| CoDA [7] | 6.71 | 38.72 | 13.66 | 6.54 | 21.57 | 9.04 |
| INHA [74] | 8.91 | 42.17 | 16.18 | 7.79 | 25.1 | 10.68 |
| CoDAv2 [75] | 9.17 | 42.04 | 16.31 | 9.12 | 23.35 | 11.49 |
| OV-Uni3DETR [9] | 12.96 | 49.25 | 20.85 | 15.21 | 31.86 | 17.99 |
| GLRD [76] | 12.96 | 49.40 | 20.88 | 17.29 | 26.78 | 18.87 |
| **OP3Det (ours)** | **14.31** | **49.63** | **21.99** | **17.77** | **32.12** | **20.16** |

### 4.3 Outdoor 3D Detection Generalization

We then evaluate OP3Det on the outdoor KITTI dataset, and list the comparative results in Tab. 3. Unlike indoor point clouds, outdoor point clouds are usually collected by the LiDAR sensor. In outdoor 3D scenes, foreground objects are usually small and sparse, with significantly fewer points. Background points dominate the scene thus disturbing the detection process significantly. The gap between outdoor LiDAR points and 2D images is thus larger than indoor ones, making leveraging 2D semantic knowledge more challenging in outdoor scenes. Additionally, since the counts of pedestrian and cyclist classes are considerably lower than that of cars, the detection AP for novel classes has only a limited effect on the overall AP. Despite this, OP3Det still achieves the best performance. Notably, on the medium difficulty level, the most important metric, $AP_{3D}$ outperforms existing methods by more than 1%, with consistent improvements also observed in $AP_{BEV}$. This largely underscores the generalization of our approach across diverse point cloud scenes and highlights the adaptability of our method. The universality of OP3Det for exploring 2D semantic knowledge is thus further validated.

### 4.4 Class-Specific 3D Detection Generalization

The ability of class-agnostic object detection to learn open-world objectness and thereby locate all objects in 3D scenes makes it highly valuable for a wide range of downstream tasks. To further demonstrate the strength of this capability, we extend OP3Det to class-specific 3D detection in this section. Specifically, we replace the 2D class-agnostic model with a class-specific detector [30], enabling the assignment of category labels during the 3D object discovery process. We follow the experimental setup of CoDA [7] and compare our approach with open-vocabulary methods, presenting the results in Tab. 4. As shown, under the class-specific setting, our method outperforms OV-Uni3DETR by more than 2% in $AP_{novel}$, and also surpasses the current state-of-the-art method, GLRD. This further validates the strong capability and practical value of our approach.

Although OP3Det is designed in a class-agnostic setting, it still performs well on class-specific detection tasks. This can be attributed to its strong 3D objectness learning, which provides high object recall and precise localization. When coupled with a class-specific head, the model readily adapts to semantic recognition, demonstrating that robust objectness understanding serves as a transferable foundation for both class-agnostic and class-specific 3D detection.

### 4.5 Ablation Study

We conduct an ablation study to evaluate the effectiveness of the SAM, multi-scale point sampling (PS), and cross-modal MoE (CM-MoE). Such a study is listed in Tab. 5 and Tab. 6.

Table 5: **Ablation Study on the SUN RGB-D dataset.** SAM: utilizing SAM for object discovery. PS: multi-scale point sampling in 3D novel object discovery. CM-MoE: cross-modal MoE.

| SAM | PS | CM-MoE | $AR_{novel}$ | $AR_{all}$ | $AR_{base}$ |
|---|---|---|---|---|---|
| | | | 54.2 | 84.0 | 92.3 |
| ✓ | | | 50.0 | 74.1 | 81.6 |
| ✓ | ✓ | | 69.2 | 87.9 | 92.5 |
| ✓ | ✓ | ✓ | **78.8** | **89.7** | **93.1** |

Table 6: **Ablation Study on the SUN RGB-D dataset about the cross-modal MoE.** PC and Img indicate whether the point cloud or image modalities are used during training, and method denotes the multi-modal fusion approach. Addition and Concat represent feature summation and concatenation, respectively, while CM-MoE refers to the proposed cross-modal Mixture of Experts.

| PC | Img | method | $AR_{novel}$ | $AR_{all}$ | $AR_{base}$ |
|----|-----|--------|--------------|------------|-------------|
| ✓ |   | - | 69.2 | 87.9 | 92.5 |
|   | ✓ | - | 38.4 | 64.4 | 72.5 |
| ✓ | ✓ | addition | 65.4 | 85.6 | 91.4 |
| ✓ | ✓ | concatenation | 66.0 | 85.8 | 92.1 |
| ✓ | ✓ | CM-MoE | **78.8** | **89.7** | **93.1** |

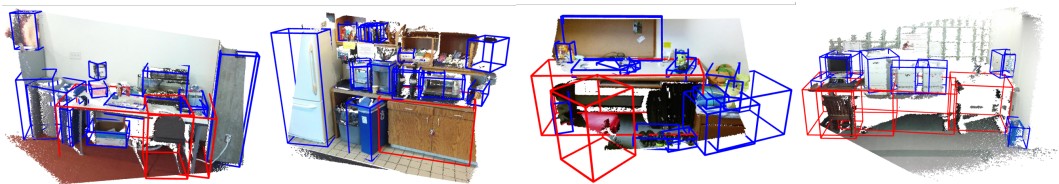

Figure 4: **The visualized results of OP3Det** on the SUN RGB-D (the first row) and ScanNet (the second row) dataset. The red boxes are base classes and blue boxes are novel classes.

The core design of our method consists of two main components: 3D novel object discovery and a cross-modal Mixture of Experts (MoE). For 3D object discovery, we first employ the robust SAM model to identify potential objects. However, using SAM alone leads to suboptimal performance on both novel and base categories, resulting in a 4.2% decrease in $AR_{novel}$ and a 10.7% decrease in $AR_{base}$. This is primarily because SAM is not inherently object-centric and tends to produce numerous fragmented or partial masks, introducing substantial noise that degrades overall detection performance. After incorporating our multi-scale point sampling strategy and a class-agnostic 2D detector for post-processing, the 3D objectness learning is significantly enhanced — $AR_{novel}$ improves by 19.2% and $AR_{base}$ by 10.9%. These results demonstrate that our multi-scale point sampling effectively suppresses noisy masks and leverages 2D semantic cues to accurately discover 3D objects. Additional results and analyses are provided in the Appendix D.

As can be seen in Tab. 6, both single-modal point cloud and RGB image modalities achieve commendable 3D objectness learning, validating the effectiveness of our strategy of using 3D spatial proximity for 3D object discovery. However, naive fusion approaches such as summation or concatenation—as commonly used in prior works—lead to degraded performance compared to the point-cloud-only model. This stems from the open-world class-agnostic setting, where binary foreground–background prediction can cause RGB features to interfere with critical 3D geometric cues if fusion is not properly guided. In contrast, our cross-modal MoE dynamically balances uni-modal and multi-modal representations, allowing each modality to contribute adaptively. As a result, OP3Det improves $AR_{all}$ by 1.8% and $AR_{novel}$ by 9.6%, effectively leveraging complementary 2D semantic and 3D geometric information while preserving modality-specific knowledge for robust 3D objectness learning.

**Visualization.** We provide visualization in Fig. 4 to further validate the effectiveness of our OP3Det.

## 5 Conclusion

We introduce OP3Det, the first work to learn 3D objectness in a class-agnostic manner in an open-world setting. We leverage multi-modal learning to bridge the gap between limited 3D annotations and extensive 2D semantic knowledge. First, we utilize 2D and 3D object priors for 3D object discovery. By integrating semantic knowledge and a cross-modal mixture of experts, OP3Det captures intra- and inter-modal dependencies cohesively and demonstrates impressive generalization capability across a diverse range of categories and scenes, especially unseen classes. Extensive experiments demonstrate its discover-*all* ability. We believe OP3Det represents a significant step forward in enabling scalable, real-world applications of 3D object detection in complex, open-world settings.

# 6 Acknowledge

The authors would like to thank anonymous reviewers for their insightful comments and valuable suggestions. This work is partially supported by the National Science Foundation under Grant No. 2047822, 1952096, and 2411151.

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

## A    Overview of 3D Open-World Learning Research

Table 7: **The overview of existing detectors on their capability.** For category, we discuss whether the detector can recognize novel classes during inference ("closed" *v.s.* "open"), and "open (no cfs)" denotes whether the category confusion problem exists in the large-vocabulary scene.

| Method | Venue | Task | | Scene (3D) | | Modality (3D) | | Category | | |
|---|---|---|---|---|---|---|---|---|---|---|
| | | 2D | 3D | indoor | outdoor | PC | Img | closed | open | open (no cfs) |
| DETR [58] | ECCV'20 | ✓ | ✗ | - | - | - | - | ✓ | ✗ | ✗ |
| DINO [77] | ICLR'23 | ✓ | ✗ | - | - | - | - | ✓ | ✗ | ✗ |
| ViLD [78] | ICLR'22 | ✓ | ✗ | - | - | - | - | | ✓ | ✗ |
| OV-DETR [79] | ECCV'22 | ✓ | ✗ | - | - | - | - | ✓ | ✓ | ✗ |
| Detic [28] | ECCV'22 | ✓ | ✗ | - | - | - | - | ✓ | ✓ | ✗ |
| ORE [80] | CVPR'21 | ✓ | ✗ | - | - | - | - | ✓ | ✓ | ✓ |
| LDET [55] | ECCV'22 | ✓ | ✗ | - | - | - | - | ✓ | ✓ | ✓ |
| UniDetector [29] | CVPR'23 | ✓ | ✗ | - | - | - | - | ✓ | ✓ | ✓ |
| VoteNet [16] | ICCV'19 | ✗ | ✓ | ✓ | ✗ | ✓ | ✗ | ✓ | ✗ | ✗ |
| FCAF3D [17] | ECCV'22 | ✗ | ✓ | ✓ | ✗ | ✓ | ✗ | ✓ | ✗ | ✗ |
| NeRF-Det [81] | ICCV'23 | ✗ | ✓ | ✓ | ✗ | ✗ | ✓ | ✓ | ✗ | ✗ |
| PointPillars [70] | CVPR'19 | ✗ | ✓ | ✗ | ✓ | ✓ | ✗ | ✓ | ✗ | ✗ |
| CenterPoint [18] | CVPR'21 | ✗ | ✓ | ✗ | ✓ | ✓ | ✗ | ✓ | ✗ | ✗ |
| BEVFormer [82] | ECCV'22 | ✗ | ✓ | ✗ | ✓ | ✗ | ✓ | ✓ | ✗ | ✗ |
| ImVoteNet [66] | CVPR'20 | ✗ | ✓ | ✓ | ✗ | ✓ | ✓ | ✓ | ✗ | ✗ |
| TR3D [56] | ICIP'23 | ✗ | ✓ | ✓ | ✗ | ✓ | ✓ | ✓ | ✗ | ✗ |
| MetaBEV [83] | ICCV'23 | ✗ | ✓ | ✗ | ✓ | ✓ | ✓ | ✓ | ✗ | ✗ |
| ImVoxelNet [84] | WACV'22 | ✗ | ✓ | ✓ | ✓ | ✗ | ✓ | ✓ | ✗ | ✗ |
| Cude RCNN [85] | CVPR'23 | ✗ | ✓ | ✓ | ✓ | ✗ | ✓ | ✓ | ✗ | ✗ |
| Uni3DETR [15] | NeurIPS'23 | ✗ | ✓ | ✓ | ✓ | ✓ | ✗ | ✓ | ✗ | ✗ |
| OV-3DET [86] | CVPR'23 | ✗ | ✓ | ✓ | ✗ | ✓ | ✗ | ✓ | ✓ | ✗ |
| CoDA [7] | NeurIPS'23 | ✗ | ✓ | ✓ | ✗ | ✓ | ✗ | ✓ | ✓ | ✗ |
| OV-Uni3DETR [9] | ECCV'24 | ✗ | ✓ | ✓ | ✓ | ✓ | ✓ | ✓ | ✓ | ✗ |
| ★ OP3Det (ours) | | ✓ | ✓ | ✓ | ✓ | ✓ | ✓ | ✓ | ✓ | ✓ |

We list the overview of existing object detectors about their capability in Tab. 7. In the 2D detection area, significant advancements have been achieved across various approaches, including traditional closed-world detectors, open-vocabulary detectors constrained by category confusion issues, and open-world detectors utilizing class-agnostic classification. However, in the 3D domain, progress remains substantially lagging behind the rapid developments observed in 2D detection. The majority of 3D detectors are designed to operate in either indoor or outdoor point clouds, lacking the ability to generalize across different environments. In terms of modality, most 3D detectors are limited to utilizing only one type of data, either point clouds or RGB images, and are constrained to the closed-world setting. While recent advancements have introduced open-vocabulary 3D detection methods, class-agnostic open-world 3D detectors overcoming the category confusion problem have still yet to emerge.

In comparison, our OP3Det, as the first class-agnostic open-world 3D detector, can not only recognize both base and novel classes during inference but also effectively mitigate the issue of category confusion. Furthermore, it leverages data from multiple modalities for multimodal training and is capable of functioning seamlessly in both indoor and outdoor scenes. Additionally, our method can be easily extended to 2D detection tasks, demonstrating its versatility and robustness. Therefore, it greatly advances existing research towards the goal of universal 3D object detection and we believe OP3Det can become a significant step towards the future of 3D foundation models.

## B    More Method Details

We summarize our method in Algorithm 1 and 2. Specifically, OP3Det utilizes both point clouds and RGB images for multi-modal training to detect in the 3D open world. To recognize novel classes in the open world and achieve the cross-category ability, the core idea of our method is to leverage abundant 2D semantic knowledge to enhance 3D open-world detection. Specifically, we utilize 3D spatial proximities to refine the uniformly distributed point prompts provided to SAM. Instead of uniformly sampling a 64×64 grid, we assign a point-wise object prior to each point by combining its

---

**Algorithm 1** - 3D object discovery algorithm

---

**Input:**
1. $(X_P, X_I)$: point clouds and corresponding 3D detection images, with camera parameters $K$, $R_t$
2. 3D annotations $\{(c_i, bb_i^{3D})\}_{i=1}^M$, $c_i = \{0, 1\}$ due to the class-agnostic setting.
2. The pre-trained foundation model SAM $\Phi_{SAM}$, a pre-trained 2D class-agnostic detector $\Phi_{CA}$
3. Point number threshold $N_{point}$ and multi-scale threshold $\{\delta_i, i = 1, 2, \cdots, N_\delta\}$

**3D Object Discovery**:
   1. Multi-scale point sampling:
     **for** $\ell = 1$ to $N_\delta$ **do**
       Initialize selected set $S_\ell \leftarrow \emptyset$, extract object prior map $O_{\text{prior}}$
       **while** $|S_\ell| < N_{point}$ **do**
         Select point $s^*$ with the highest value in $O_{\text{prior}}$, add $s^*$ to $S_\ell$
         **for all** $p_i \in X_p \setminus S_\ell$ **do**
           Compute 3D distance $\texttt{d}(\texttt{s}^*, \texttt{p}_\texttt{i})$
           **if** $\texttt{d}(\texttt{s}^*, \texttt{p}_\texttt{i}) < \delta_\ell$ **then**
             Set object prior $O_{\text{prior}}$ at corresponding 2D pixel to 0
           **end if**
         **end for**
       **end while**
     **end for**
     The ultimate selected set $S = \text{NMS}(\bigcup_{\ell=1}^{N_\delta} S_\ell)$
   2. Apply $\Phi_{SAM}$ and $\Phi_{CA}$ for selected points, then utilizing $K$, $R_t$ to project into the 3D space:
     $\{\hat{bb}_i\} = (KR_t)^{-1} \cdot \Phi_{CA}(\Phi_{SAM}(S))$.

---

---

**Algorithm 2** - Cross-Modal MoE algorithm

---

**Cross-Modal MoE Training**:
   1. Obtain point cloud features $F_P$ from $X_P$, image features in the voxel space $F_I'$ from $X_I$.
   2. Obtain multi-modal features $F_M = [F_P, F_I']$.
   3. Utilize self-attention module on the features:
     $\mathcal{F}_P = \text{SelfAttn}(F_P)$, and $\mathcal{F}_I$, $\mathcal{F}_M$ is obtained in the same way.
   4. Utilize the multi-modal router $\mathcal{R}$ to obtain the routing probability: $(p_P, p_I, p_M) = \mathcal{R}(\mathcal{F}_M)$.
   5. Apply the cross-modal MoE for multi-modal fusion: $\mathcal{F} = \sum\limits_{i \in (P,I,M)} p_i \cdot \mathcal{E}_i(\mathcal{F}_i)$.

   6. Utilize $\mathcal{F}$ for 3D bounding box prediction, supervised with $\{\hat{bb}_i\} + \{bb_i\}$ for the model training.

---

IoU scores with the maximum attention value across self-attention heads from the self-supervised model (DINO). These object prior points are then progressively refined using a coarse-to-fine multi-scale sampling strategy, which allows for increasingly precise localization across spatial resolutions. Specifically, we project all 3D points onto the 2D image plane and establish a mapping between 2D pixels and 3D points by associating each pixel with its nearest 3D point. In each iteration, the point with the highest object prior value is selected as the source point. For each selected source point, we compute its 3D distance to all other points. Points within a predefined 3D distance threshold are suppressed by setting their 2D object prior values to zero. We then select the next source point with the highest remaining object prior to the score and repeat this process until N points are selected. The 2D bounding boxes generated from the refined points are then used in 2D class-agnostic models for 3D object discovery.

Ultimately, during training, the cross-modal MoE is utilized for multi-modal fusion and the discovered 3D objects serve as supervision.

## C  More Experimental Results

We further conduct more experiments in this section to demonstrate the effectiveness of our designs. We first list the AP metric of OP3Det on the SUN RGB-D dataset in Tab. 8, then conduct ablation study

Table 8: **AP metric of the cross-category performance of OP3Det on the SUN RGB-D dataset.**

| Method | SUN RGB-D | | |
|---|---|---|---|
| | $AP_{novel}$ | $AP_{all}$ | $AP_{base}$ |
| *closed-world 3D object detection methods* | | | |
| VoteNet [16] | 2.0 | 55.1 | 66.3 |
| GroupFree [60] | 2.3 | 49.2 | 58.4 |
| FCAF3D [17] | 3.4 | 61.0 | 74.1 |
| Uni3DETR [15] | 2.3 | 61.3 | 74.4 |
| Tr3D [56] | 3.7 | 53.4 | 62.7 |
| *open-vocabulary 3D object detection methods* | | | |
| CoDA [7] | 9.1 | 48.2 | 57.8 |
| OV-Uni3DETR [9] | 10.2 | 57.4 | 67.8 |
| ImOV3D [61] | 8.1 | 28.3 | 35.4 |
| *class-agnostic open-world 3D object detection method* | | | |
| **OP3Det (ours)** | **12.6** | **65.4** | **75.7** |

Table 9: **Ablation Study on the SUN RGB-D dataset about the multi-scale point sampling.** We conduct 3D object discovery using the corresponding methods, and directly evaluate the AR and AP metrics of discovered 3D objects, without training the detector. PS is short for point sampling.

| method | $AR_{novel}$ | $AR_{all}$ | $AR_{base}$ | AP |
|---|---|---|---|---|
| SAM [27] | 64.0 | 55.4 | 52.8 | 6.8 |
| SAM + PS ($\tau$=0.2) | 47.5 | 43.1 | 41.7 | 5.9 |
| SAM + PS ($\tau$=2) | 49.9 | 12.6 | 40.3 | 5.7 |
| SAM + multi-scale PS | 61.9 | 54.2 | 51.9 | 7.6 |
| SAM + multi-scale PS + LDET [55] | **66.1** | **59.2** | **57.1** | **10.0** |

mainly on the multi-scale point sampling strategy during 3D object discovery and the cross-modal MoE.

**AP metric.** In the original paper, we mainly report the AR metric of our OP3Det, consisting of $AR_{all}$, $AR_{novel}$ and $AR_{base}$. We only report $AP_{all}$ in the original paper. The main reason is that we aim to discover "all" 3D objects in the scene, while not all bounding boxes are necessarily annotated in the ground truth of the test set. As a result, objects that are not in the test set annotation but found by the model will also be counted as false positives (FP) and introduce errors in AP metrics. Besides, calculating $AP_{novel}$ and $AP_{base}$ is also not suitable for class-agnostic detection, because the definition of FP can be ambiguous. For example, when calculating $AP_{novel}$, it is unclear whether the detected base objects should be FP. Therefore, the AR metric is more suitable for our setting.

Despite this, we can still ignore base or novel objects when calculating $AP_{novel}$ or $AP_{base}$, to provide a comprehensive comparison. We list the results in Tab. 8. As can be seen, our OP3Det also obtains a better performance, achieving the 12.6% $AP_{novel}$ and 65.4% AP. Compared with existing methods, we achieve 2.4% higher $AP_{novel}$ and 4.1% higher AP. This further demonstrates the effectiveness of our method.

**Multi-scale point sampling.** In our original paper, for 3D object discovery, we first extract class-agnostic masks using SAM. During this process, the multi-scale point sampling strategy is employed to alleviate fragmented masks or object parts. Finally, LDET is applied for post-processing. We analyze the impact of these design choices sequentially, and evaluate the AR and AP metrics of discovered 3D objects, as shown in Tab. 9.

As observed, the direct results from SAM achieve relatively strong AR metrics for 3D discovered objects, with an $AR_{novel}$ of 64% and an $AR_{all}$ of 55.1%. This demonstrates that SAM effectively uncovers a broader range of objects. With these objects participating in the training, the diversity of training can be boosted greatly. This validates the effectiveness of our idea to introduce broader 2D semantic knowledge into the 3D domain. However, the presence of numerous fragmented masks introduces a significant amount of noisy masks, leading to a very low AP of only 6.8%. This low precision indicates that directly using these objects during training would result in poor model performance, as can be seen in our original paper.

Table 10: **Comparison with 2D open-world methods for the COCO (VOC) to COCO (non-VOC) setting.** Here, we compare with various 2D open-world instance segmentation methods and report metrics based on masks.

| Method | AP | $AR_{100}$ | $F_1$ |
|---|---|---|---|
| Mask R-CNN [87] | 1.0 | 8.2 | 1.8 |
| SAM [27] | 3.6 | 48.1 | 6.7 |
| OLN [88] | 4.2 | 28.4 | 7.3 |
| LDET [55] | 4.3 | 24.8 | 7.3 |
| GGN [89] | 4.9 | 28.3 | 8.4 |
| SWORD [90] | 4.8 | 30.2 | 8.3 |
| UDOS [91] | 2.9 | 34.3 | 5.3 |
| SOS [92] | 8.9 | 39.3 | 14.5 |
| **OP3Det (ours)** | **13.9** | **42.9** | **21.0** |

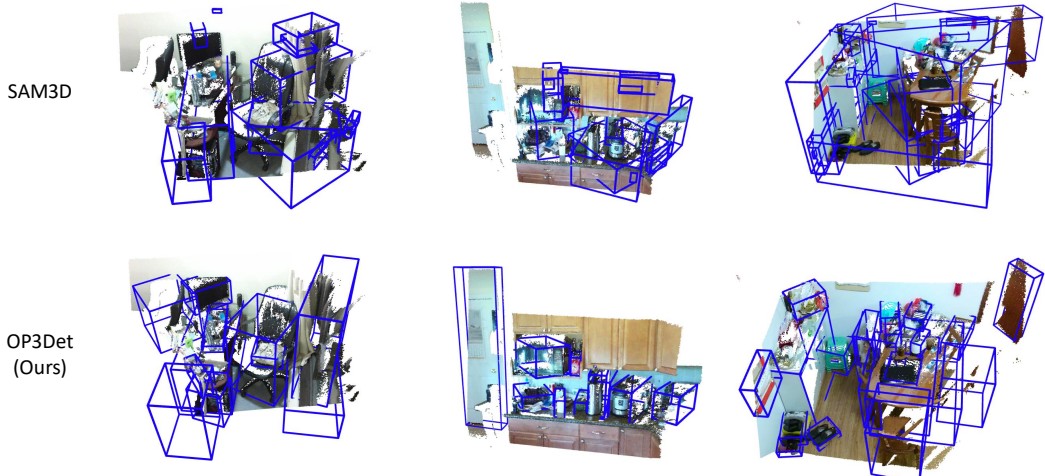

SAM3D

OP3Det
(Ours)

Figure 5: **Qualitative comparison** with SAM3D [46]. Benefiting from our contributions, our method OP3Det yields more accurate localization and precise discovery of novel objects.

Using point sampling effectively filters out many fragmented masks, reducing noise in object masks. However, this also inadvertently filters some useful objects. Consequently, regardless of the choice of $\tau$, both AR and AP metrics show a decline, negatively impacting the performance. By leveraging multi-scale point sampling that combines the strengths of different $\tau$ values, it becomes possible to balance noise reduction while retaining diverse objects. This results in an improved AP of 7.6%, demonstrating a notable enhancement in the quality of discovered 3D objects. However, the AR metric still remains lower than when directly using SAM, indicating that some important masks are still being filtered out. By further incorporating LDET, the holistic object understanding ability of the 2D detector can be utilized to better filter out noise within object masks. As a result, both AR and AP metrics show significant improvement. This enhancement demonstrates that the quality of discovered 3D objects is notably elevated, enabling their effective use in subsequent training and ensuring the model's cross-category generalization ability.

**Comparison with 2D methods.** Additionally, since a part of our method is conducted in the 2D domain, we also compare it with 2D open-world detectors. Specifically, we first perform 2D object discovery and then train Mask R-CNN [87] on the COCO [31] dataset, for instance, segmentation. The 20 classes overlapping with VOC [93] are treated as seen classes, while the remaining 60 classes are treated as novel ones. We compare the predicted masks against existing methods with the results presented in Tab. 10. For a fair comparison, we do not utilize the text prompts here, as utilizing the class-specific 2D detector may result in the category leakage problem. As can be seen, OP3Det also surpasses existing methods by 5% in AP and 3.6% in AR. This demonstrates the effectiveness of our designed multi-scale point prompts in 2D object discovery.

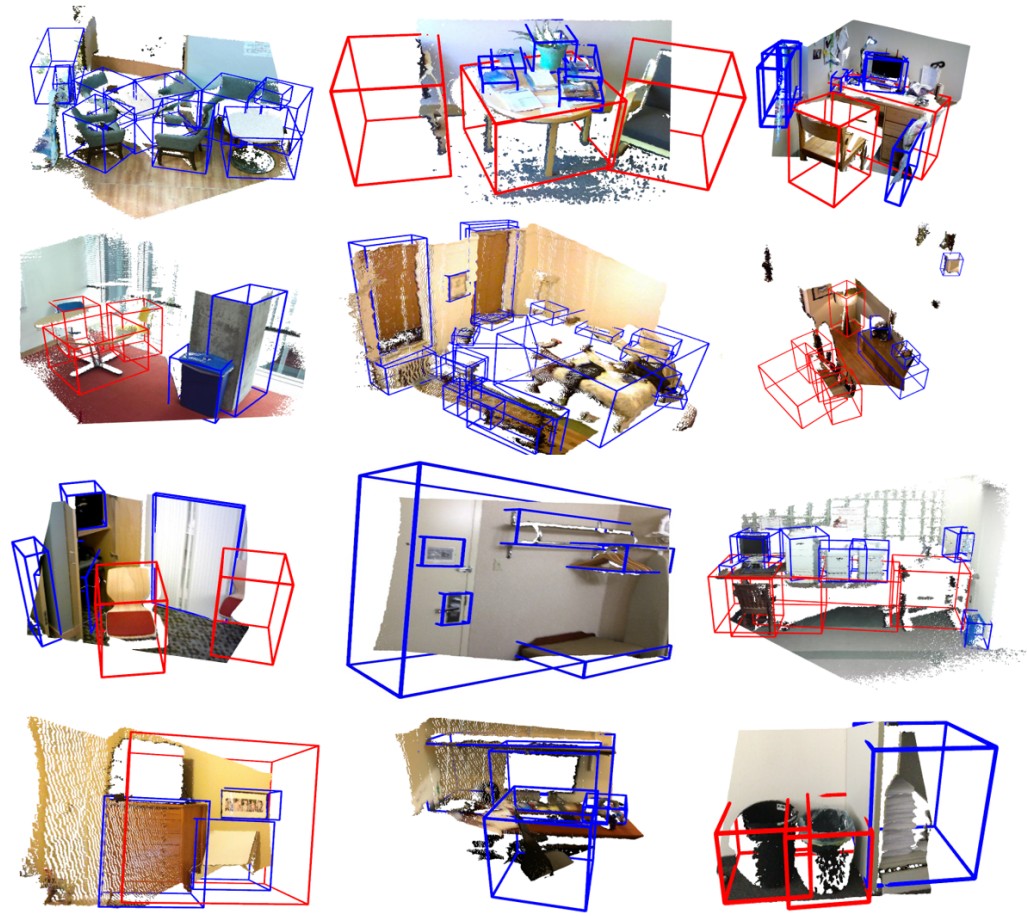

Figure 6: **More visualized results of OP3Det** on the SUN RGB-D (the first three rows) and ScanNet (the last row) dataset. The red boxes are base classes and blue boxes are novel classes.

**Comparison with SAM-related methods.** Recent approaches such as SAM3D [46] leverage SAM-generated masks from RGB images and project them into 3D to obtain class-agnostic instance segmentation. To improve mask consistency, SAM3D adopts a bidirectional merging strategy across adjacent frames. However, it does not perform object detection. The instance segmentation task requires more fine-grained mask annotations for training; thus utilizing geometric information can be easier in such a setting. In comparison, the class-agnostic object detection task has still not been explored yet. Meanwhile, existing SAM-related methods usually rely heavily on temporal fusion for filtering low-quality objects and achieving better localization quality, which cannot be applied in our object detection setting, where only one frame is available for a 3D scene. To ensure a fair comparison under favorable conditions for SAM3D, we re-implement its pipeline and extract 3D bounding boxes around the resulting point cloud masks. As shown in Figure 5, our method OP3Det outperforms SAM3D significantly in both the number and localization quality of novel objects. This demonstrates that OP3Det is not only more effective in discovering novel instances, but also robust under minimal inputs—achieving superior results without requiring multi-frame fusion.

## D    More Visualized Results

We provide more visualized results on the SUN RGB-D and ScanNet datasets in Fig. 6. From these examples, it is evident that our model effectively detects abundant base class objects, such as chairs and tables, while also accurately identifying numerous rare novel classes, such as various small objects on tables or beds. These visualizations further validate the effectiveness of our approach.

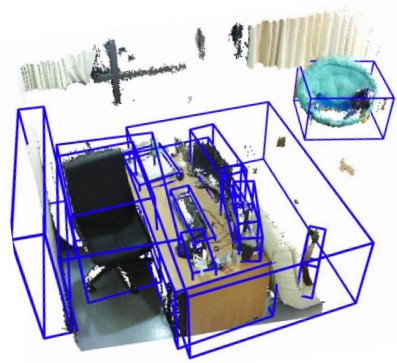

Figure 7: **Failure Case.** Despite accurately detecting most objects using both 2D semantic and 3D geometry knowledge, OP3Det fails on non-rigid and low-contrast regions such as the white curtains.

**Limitation.** All methods have the potential for errors, and here we discuss the potential failure cases in our results. As observed in the visualization results, OP3Det successfully detects most objects, regardless of their size or whether their category can be clearly identified. However, some objects remain undetected, particularly in complex scenes, such as paper items on a cluttered desk or stickers on a wall. This is partially because the training data and annotations do not cover a sufficiently diverse range of scenarios. Additionally, in highly complex environments, missed detections may occur when objects have rigidity and color similar to the background, making them difficult to distinguish.

**Failure Case Analysis.** As shown in the figure7, our model successfully detects most objects in this complex scene. The two closely black monitors on the desk are correctly localized, benefiting from the use of 3D geometric information. However, our method still fails to detect some objects such as the white curtain and the white sofa. We attribute these failures to the lack of distinctive geometric or color features in these objects. Since both curtains and sofas have relatively flat geometry and low texture contrast, especially under overexposed lighting conditions, it becomes difficult for the model to distinguish them from the background or surrounding surfaces. This suggests the need for better handling of low-texture, non-rigid and color-homogeneous regions in open-world 3D detection.

