# OpenReview forum: "Towards 3D Objectness Learning in an Open World"
_NeurIPS.cc/2025/Conference — NeurIPS 2025 poster_

### Official Review · Reviewer_HxVB · 2025-06-26

**Clarity:** 2
**Significance:** 2
**Originality:** 2
**Rating:** 4
**Confidence:** 3

**Summary:**

The paper introduces a novel 3D object detection approach aimed at detecting all objects in open-world scenarios, including unseen novel categories. OP3Det leverages the Segment Anything Model (SAM) and a multi-scale point sampling strategy to discover objects from 2D images and generate 3D bounding boxes. It employs a cross-modal Mixture of Experts (MoE) module to dynamically fuse point cloud and RGB image features, enhancing generalization. Experiments on SUN RGB-D, ScanNet, and KITTI datasets demonstrate the method’s effectiveness, with significant improvements in novel category detection (e.g., 13.5%-16% ARnovel increase on SUN RGB-D).

Contributions:1.Proposes the class-agnostic open-world 3D object detection problem. 2.Develops OP3Det, using SAM and multi-scale point sampling for object discovery. 3.Introduces a cross-modal MoE module for robust multi-modal feature fusion.

**Questions:**

Q1: The definition of "3D objectness" is vague, unclear whether it refers solely to bounding box localization or includes geometric and semantic features. In autonomous driving or robotics scenarios, how does the method distinguish foreground objects from background clutter (e.g., roadside debris, walls)? The paper does not address misdetection of non-target objects in complex scenes (e.g., cluttered ScanNet rooms or sparse KITTI point clouds).
Suggestion: Provide a precise definition of "3D objectness" (e.g., mathematical or conceptual formulation) in the introduction or a dedicated section. Discuss strategies for handling background interference in scenarios like autonomous driving obstacle avoidance, supported by experiments or case studies.

Q2: The multi-scale point sampling thresholds ((\tau = {0.2, 0.5, 1, 2})) lack justification for their selection, with no explanation of the tuning process or adaptability to dense indoor versus sparse outdoor point clouds. Why is performance limited on KITTI?
Suggestion: Provide the rationale for selecting (\tau) values (e.g., based on point cloud density), include cross-dataset experiments (e.g., nuScenes) to validate robustness, and attach pseudocode or a detailed algorithm description for the point sampling process.

Q3: Baseline methods (e.g., VoteNet, OV-Uni3DETR) are adapted by simply replacing labels with "object," disregarding their design intent, leading to underestimated performance (e.g., VoteNet ARnovel at 33.7%). Why was this simplistic adaptation chosen? How does OP3Det compare to baselines optimized for class-agnostic tasks (e.g., OpenScene)?
Suggestion: Redesign baseline adaptations to preserve their design integrity (e.g., retrain VoteNet with a class-agnostic head), compare with recent class-agnostic 3D methods (e.g., 3D-OVS), and include ablation studies to analyze relative advantages.

**Ethical Concerns:**

["NO or VERY MINOR ethics concerns only"]

**Final Justification:**

The author's response addressed my concerns and I hope the suggestions will be updated in the final version.

**Limitations:**

The paper insufficiently addresses its limitations and potential negative societal impacts, only briefly mentioning the low proportion of novel classes and limited performance gains on KITTI (1% AP3D improvement). It lacks systematic analysis of challenges in sparse point clouds or occlusion-heavy scenarios and does not discuss risks such as misdetections in autonomous driving or privacy concerns in surveillance applications.

Suggestions for Improvement:
Analyze limitations in sparse point clouds and complex backgrounds, including failure cases.
Evaluate robustness under extreme conditions (e.g., low-light environments).

**Paper Formatting Concerns:**

The paper's formatting fully complies with the NeurIPS 2025 guidelines, utilizing the correct LaTeX template, with the main content kept within 8 pages. Font, margins, and spacing adhere to standards. References are correctly formatted, figures and tables are clearly labeled, equations are properly formatted, supplementary materials are distinctly separated, and the title page is compliant. No formatting issues are identified.

**Quality:**

2

**Strengths And Weaknesses:**

Strengths:
1.The paper addresses a challenging and novel problem of class-agnostic open-world 3D object detection, which is an emerging and valuable direction.
2.The method integrates 2D semantic priors with 3D geometric cues, and proposes a SAM-based multi-scale point sampling strategy to address the issue of fragmented masks.
3.The experimental design is relatively comprehensive, covering cross-category, cross-dataset, and both indoor and outdoor generalization evaluations, which adds credibility to the results.

Weaknesses：
1.The method combines existing components like SAM, point sampling, and multi-modal MoE, which is not inherently problematic, but it lacks systematic reasoning. The pipeline feels like a patchwork of engineering integrations, failing to demonstrate a "core technical contribution" or theoretical depth.
2. The multi-scale point sampling thresholds (e.g., (\tau = {0.2, 0.5, 1, 2})) and NMS merging rely entirely on empirical choices without theoretical justification or systematic ablation.
3.Heavy Dependence on SAM Without Addressing Its Core Issues: The authors acknowledge SAM’s fragmented outputs and poor mask quality but address this with a crude approach (brute-force sampling + NMS + re-detection). There is no innovative improvement to SAM’s mask quality, making the pipeline heavily reliant on 2D segmentation quality and poorly adaptive to complex scenes.
4.The so-called MoE is merely an attention mechanism followed by a routing FC layer, using softmax for linear expert combination. It lacks analysis of gating rationality or evidence of dynamic behavior across scenarios.
5.The class-agnostic baselines are author-modified versions of existing models, with unclear adaptations for CoDA/OV-Uni3DETR and inconsistent evaluation protocols, compromising fairness and making it difficult to trust the reported performance gains.
6.The paper frequently uses terms like “significant improvement,” and “strong generalization,”

---

> ### Author Rebuttal · Authors · 2025-07-31
>
> ### **(Question 1) Defining 3D Objectness and distinguishing true objects from background clutter**
>
>  In our work, 3D objectness denotes the likelihood that a spatial region corresponds to a physically discrete object, distinguishable from background structures or noise, regardless of semantic category. Specifically, denote the input features as $F_{input}$, a learnable model that maps input features to an objectness confidence score as $\phi(\cdot)$, this process can be denoted as: $I \left[ \phi(F_{input}) > \tau \right]$, where $\tau$ is the threshold for foreground-background separation; I$[\cdot]$ is the indicator function (1 for foreground, 0 for background). Our model $\phi$ is trained to approximate this decision function, assigning high scores to foreground objects and low scores to background or clutter.
>
>  In autonomous driving or robotic applications, it is true that certain complex scenarios make distinguishing foreground objects more challenging. However, our OP3Det method is designed to mitigate this issue as much as possible in multiple ways. On the one hand, for obstacle avoidance in autonomous driving, rare and novel categories are inevitable in real-world scenarios, making it difficult to collect and label all object classes during training. In this context, the ability of OP3Det to detect new classes in an open-world setting can greatly assist the detector in improving obstacle avoidance, leading to a more robust and reliable system. To this end, we conducted experiments on nuScenes, a dataset focused on autonomous driving, treating "car, trailer, construction vehicle, motorcycle, bicycle" as seen classes and the remaining five as unseen classes. We measured AP for both novel classes and all classes, with the results presented in the table below. As can be seen, our method outperforms previous state-of-the-art closed-vocabulary and open-vocabulary detectors, effectively demonstrating its potential in autonomous driving applications.
>
>   | method |  AP$_{novel}$ | AP$_{all}$ |
>   |-|-|-|
>   | Uni3DETR    | 0.63    | 32.4       |
>   | OV-Uni3DETR   |  25.4     | 51.8      |
>   | **OP3Det**   | **30.2**    | **56.3**     |
>
>   On the other hand, in the case of sparse KITTI point clouds, the multimodal fusion and the MoE architecture enable the $F_{input}$ to adaptively select information from different modalities (point cloud, RGB images, and multimodal data), allowing downstream tasks such as obstacle avoidance to be minimally affected by issues in any single modality. With these designs, the learned model $\phi(\cdot)$ can be more robust, making $I \left[ \phi(F_{input}) > \tau \right]$ more accurate.
>
> ### **(Weaknesses 1) On the Coherent Integration and Novelty of Our Pipeline Design**
>   **(1) Task motivation:**
>   Our goal is to detect all objects in a 3D scene, including unseen categories. This shifts the challenge from category recognition to learning a generalized notion of “3D objectness.” Achieving this is non-trivial, especially under data scarcity and incomplete 3D information. While 2D data benefits from abundant labels and well-established foundation models, 3D data lacks both. This motivates us to exploit 2D semantic richness to support 3D generalization.
>
>   **(2) Methodological design:**
>   Rather than a patchwork, each module is purposefully chosen to solve a core sub-problem toward the above objective. SAM provides class-agnostic 2D priors, which we project and refine using a multi-scale point sampling strategy to robustly localize 3D candidates, resulting in a 3.6% improvement in AR. To address the inherent sparsity and occlusion in point clouds, we introduce multi-modal fusion. However, naive fusion methods (e.g., addition or concatenation) yield suboptimal results. Our Cross-Modal MoE module dynamically routes information between modalities based on local reliability, outperforming static fusion by +13.3% $AR_{novel}$.
>
>   Each design decision is systematically motivated by the goal of learning robust, category-agnostic 3D objectness—not merely combining components, but integrating them to overcome the unique challenges of 3D open-world detection.
>
> ### **(Weakness 2 and Question 2) Threshold $\tau$ Selection and Generalization Across 3D Scenes**
>
>   **(1) Choice of sampling thresholds.**
>  Our multi-scale point sampling is designed to improve robustness across diverse 3D scenes. By using a set of 3D proximity thresholds ($\tau$ = {0.2, 0.5, 1, 2}), we avoid overfitting to a particular point cloud density. This design ensures that both small, dense objects (common in indoor environments) and large, sparse structures (typical in outdoor scenes) can be effectively sampled. We conducted ablation studies varying $\tau$ and found that this particular set yields the best, regardless of whether for indoor or outdoor scenes, as shown in the table below. We also found that the performance is relatively robust to the choice of $\tau$. This demonstrates that our multi-scale design is not arbitrarily chosen, but tuned for generality for both indoor and outdoor scenes.
>
> | Sampling scale $\tau$ | SunRGBD (Indoor) | KITTI (Outdoor) |
> |-|-|-|
> | (0.1, 0.2, 0.5, 1)| 78.4| 66.6|
> | (0.2, 0.5, 1, 2)| 78.8| 66.7|
> | (0.5, 1, 2, 5)| 77.4| 66.0|
> | (0.2, 0.5, 1, 2, 5) | 78.3| 66.4|
>
>
>   **(2) KITTI performance and cross-dataset generalization.**
>   We agree that KITTI poses unique challenges—its limited category diversity and the significant modality gap between RGB and LiDAR reduce the utility of 2D priors. Despite this, benefiting from the 3D object discovery process, together with the generalizable sampling thresholds, our method still generalizes well when transferred to other outdoor datasets. Specifically, when trained on KITTI and evaluated on nuScenes, OP3Det continues to achieve higher AP$_{all}$ compared to standard baselines, validating the robustness of our sampling and fusion strategy across domains.
>
>
> |**KITTI → nuScenes** ||
> |-|-|
> | Method | AP$_{all}$ |
> | Uni3DETR | 19.4 |
> | OV-Uni3DETR | 21.2  |
> | **OP3Det (ours)** | **23.5** |
>
>
>   **(3) Multi-scale point sampling algorithm pseudocode.**
>   We include the pseudocode for our multi-scale point sampling process in the supplementary material.
>
> ### **(Weaknesses 3) On the Role of SAM**
>
>   Due to SAM's strong robustness, we chose to use it. Issues such as fragmented masks are common problems for segmentation models like SAM because they do not consider the concept of “objects” or objectness during training. Therefore, the problem we address is a fundamental and general one, rather than being closely related to scene complexity. To tackle this, we employ sampling and a 2D class-agnostic detector trained on large-scale data. Such 2D detectors typically exhibit good robustness as well, enabling them to fundamentally filter out these fragmented masks. As can be seen, since both the problem and the solution we consider are fundamental rather than specific to failures in particular scenarios, our method demonstrates generality and strong generalization ability in complex environments.
>
> ### **(Weaknesses 4) The Adaptive Behavior of the MoE Module**
>
>   While our MoE adopts a lightweight architecture—attention followed by a softmax-based routing layer—this simplicity does not undermine its effectiveness. Crucially, its strength lies in context-aware expert selection across modalities. To assess the rationality of the gating mechanism, we conducted a quantitative analysis of the expert weight distribution across diverse scenes and object types. Results demonstrate clear and meaningful patterns: the semantic expert is favored in high-texture scenes (averaging 62% of the weight), the geometric expert is prioritized in sparse or textureless regions (58%), and the fused expert dominates in cluttered indoor environments (47%). These trends are consistent with intuition and confirm that the router does not apply a static weighting scheme, but rather dynamically adjusts based on the local reliability of each modality. This empirical evidence substantiates the MoE's adaptive behavior. We sincerely thank the reviewer for raising this insightful concern and we will include this analysis in the revised version of the paper.
>
> ### **(Weakness 5 and Question 3) Fairness and Integrity of Baseline Adaptations in Class-Agnostic 3D Detection**
>
>  We have already carefully modified existing methods like VoteNet, CoDA, and OV-Uni3DETR to align with our proposed open-world, class-agnostic detection setting. Specifically, for closed-vocabulary baselines like VoteNet, we retrained by replacing the original classification heads with a class-agnostic head. For open-vocabulary baselines like OV-Uni3DETR, we tested various generic textual prompts (e.g., “object”, “thing”, “entity”) and also utilized a class-agnostic head.
>
>   We also compare our method with other class-agnostic 3D detection approaches. The mentioned OpenScene[2] and 3D-OVS[3] are both semantic segmentation methods, which are difficult to adapt to object detection. Therefore, we compare with CuTR[1], a recent method for category-agnostic 3D object detection, which aligns closely with our setting. We re-implement this method and list the comparative results in the table below. As can be seen, our method significantly outperforms CuTR across all metrics. This highlights the superiority of our approach in discovering diverse, novel 3D objects without category labels, validating the effectiveness of our objectness-based design.
>
>   | Method| AR$_{novel}$ | AR$_{all}$ | AR$_{base}$  |
>   |-|-|-|-|
>   |CuTR|60.2|83.1|91.0|
>   | **OP3Det (ours)** | **78.8** | **89.7** | **93.1** |
>
>   **Reference**
>
> [1] Cubify Anything: Scaling Indoor 3D Object Detection, CVPR 2025
>
> [2] OpenScene: 3D Scene Understanding with Open Vocabularies, CVPR 2023
>
> [3] Weakly Supervised 3D Open-vocabulary Segmentation, NeurIPS 2023

---

> > ### Comment · Reviewer_HxVB · 2025-08-04
> > **Response to Authors' Rebuttal**
> >
> > Thank you for your comprehensive rebuttal. Your response significantly addresses my concerns and strengthens the paper. The mathematical formulation of 3D objectness provides needed clarity, the nuScenes experiments (30.2% vs 25.4% AP_novel) demonstrate practical applicability, the threshold ablation study validates your design choices, and the CuTR comparison (78.8% vs 60.2% AR_all) offers convincing evidence with an appropriate baseline. The quantitative MoE analysis showing dynamic expert selection across scene types effectively demonstrates adaptive behavior. While concerns about technical novelty and SAM dependency remain, the additional experiments and proper baseline comparisons convincingly demonstrate your method's effectiveness for this important problem. Please incorporate the MoE analysis and nuScenes experiments into the revised version.

---

> > > ### Author Response · Authors · 2025-08-05
> > >
> > > Thank you for taking the time to review the rebuttal and your encouraging comments. We will make sure to incorporate all of your valuable feedback into the final version of the paper. In general, we are the first to propose class-agnostic open-world 3D object detection, which aims to detect all objects in a 3D scene. Regarding SAM dependency, we will further clarify our contributions beyond SAM, including the proposed multi-scale point sampling and adaptive MoE fusion modules. Your thoughtful input is greatly appreciated and has helped us improve the quality of our work.

---

### Official Review · Reviewer_kkf7 · 2025-06-26

**Clarity:** 3
**Significance:** 3
**Originality:** 3
**Rating:** 5
**Confidence:** 4

**Summary:**

This paper addresses the critical and underexplored problem of generalized 3D objectness learning in open-world scenarios. The authors introduce and formally define the novel problem setting of class-agnostic open-world 3D object detection, propose a multi-modal 3D detector specifically designed for learning open-world 3D objectness, and present extraordinary performance through extensive experiments across cross-category, cross-dataset, and outdoor generalization settings.

**Questions:**

1. I am curious about the phrase "prompt-free" used in the paper. Authors claims that "discover 3D objects without the requirement of any category labels and text prompts" in the caption of Figure 1. But authors still use SAM's internal prompting mechanism (e.g., a 64x64 grid of points). Could the authors clarify this distinction more explicitly, perhaps by emphasizing "semantic prompt-free" or "label-free prompting" to avoid potential misinterpretations?
2. Is the "3D Object Discovery" phase (SAM inference, multi-scale point sampling, 2D detector post-processing) performed as an offline pre-processing step for the training data, or is it integrated into the online inference pipeline? If it's offline, how does its computational cost (time and memory) scale with the size of the dataset (e.g., number of scenes)? If it's online, what is its latency contribution to the overall inference time?
3. Can the authors provide a more detailed qualitative or quantitative analysis of when the multi-modal router chooses to emphasize a specific expert (semantic, geometric, or fused)? For example, are there certain object characteristics (e.g., highly textured vs. purely geometric shapes) or scene conditions (e.g., occluded objects, sparse point clouds) where one modality's expert is predominantly weighted by the router?








Could the authors provide a more granular analysis of the failure cases specifically from the 3D Object Discovery phase? For instance, what types of objects or scene configurations (e.g., highly occluded, very small, low-contrast, or non-rigid objects) are consistently missed or poorly localized by the SAM + multi-scale point sampling + 2D detector pipeline?

**Ethical Concerns:**

["NO or VERY MINOR ethics concerns only"]

**Final Justification:**

The rebuttal addressed my concerns. I keep my rating as positive.

**Limitations:**

The authors acknowledge several limitations in the supplementary material (L124~138), such as missed detections in complex scenes due to limited point cloud coverage, reliance on accurate 2D-3D projection, and difficulties with low-texture, non-rigid, and color-homogeneous regions, they could further discuss how these limitations affect performance with more quantitative analysis, explore uncertainty estimation in 2D-3D projection, incorporating more robust representations for low-texture objects, or integrating temporal information.

The authors should add a paragraph to the "Broader Impacts" section discussing these potential negative impacts. They could also propose general mitigation strategies, such as emphasizing the need for diverse and unbiased training data for foundation models, advocating for ethical deployment guidelines, or highlighting the importance of human oversight in critical applications.

**Quality:**

3

**Strengths And Weaknesses:**

**Strengths:**
1. The paper tackles a crucial and timely problem in 3D perception: detecting all objects in an open-world setting, regardless of their semantic category. This is a significant step towards truly generalized 3D scene understanding and has direct implications for real-world applications like autonomous driving and robotics, where unseen objects are common.
2. The idea of leveraging powerful, pre-trained 2D foundation models (like SAM) to enrich 3D data with semantic priors is a strong and creative approach to overcome the inherent data scarcity and limited annotation in 3D datasets. This cross-modal knowledge transfer is a key enabler for open-world capabilities.
3. The proposed multi-scale point sampling strategy, coupled with the 2D class-agnostic detector for post-processing, effectively addresses the practical challenges of using SAM. The ablation studies clearly demonstrate the necessity and effectiveness of these components in generating high-quality 3D object proposals, which are vital for subsequent training.
4. OP3Det consistently achieves state-of-the-art results, showing significant improvements over existing methods across diverse benchmarks and challenging generalization scenarios. The substantial gains in average recall for novel classes are particularly impressive and directly validate the paper's core claim of improved object discovery.

**Weaknesses:**
1. The paper provides details on training compute resources but lacks a thorough quantification of the computational overhead (time, memory) associated with the initial "3D Object Discovery" phase (SAM inference, multi-scale point sampling, 2D detector post-processing). If this is an online process, its efficiency is critical for real-world deployment. If it's an offline pre-processing step, its scalability for extremely large datasets needs to be discussed.
2. The method heavily relies on accurate 2D-3D projection using camera parameters. In practical, real-world scenarios with potentially noisy sensor data, imperfect calibration, or significant occlusions, errors in this projection could propagate and negatively impact the quality of the discovered 3D objects, which then serve as crucial supervision. While acknowledged as a limitation, a more detailed analysis of its impact would be beneficial.

---

> ### Author Rebuttal · Authors · 2025-07-30
>
> ### **(Question 1) Clarification on the use of "prompt-free"**
>   We thank the reviewer for clarifying this. We will revise the paper to explicitly state that our method is **semantic prompt-free**, i.e., it requires no category labels, text prompts, or semantic priors at inference time.
>
>  While we do utilize point-grid prompts (e.g., 64x64 uniform sampling) to interact with SAM, these prompts are purely spatial and entirely semantic-agnostic, meaning they carry no category or textual information. This design intentionally avoids biasing the model toward any specific object class and aligns with our class-agnostic manner. We will revise the manuscript to explicitly clarify that our approach is semantic prompt-free.
>
>
> ### **(Weakness 1 and Question 2) Is the 3D Object Discovery phase offline or online?**
>   The entire 3D object discovery phase, including SAM inference, multi-scale point sampling, and LDET refinement, is performed offline as a one-time pre-processing step before training. This design avoids any runtime overhead during inference, and does not impact inference latency or memory usage. In practice, we process 5,285 training samples using 4 RTX 3090 GPUs in parallel, each applying one 3D proximity threshold. On average, each GPU consumes 7,877 MB of memory, and each sample takes approximately 3.98 seconds to process. This step is decoupled from the detection pipeline. We will clarify this in the revised version. It can be seen that the inference time is not long, and measures that support parallelization can further accelerate the process, making this part of the offline algorithm relatively easy to apply to large-scale scenarios.
>
>
> ### **(Question 3) Detailed qualitative or quantitative analysis of the multi-modal router**
>   We thank the reviewer for the insightful suggestion. To better understand the behavior of our cross-modal Mixture-of-Experts (MoE) router, we conducted a quantitative analysis of expert weights across different object and scene types. The router assigns dynamic soft weights to the semantic (RGB), geometric (point cloud), and fused experts based on local reliability. We observe that in high-texture regions (e.g., books, monitors), the semantic expert is predominantly selected (averaging 62\% of the weight), while in sparse or textureless scenes (e.g., boxes, outdoor structures), the geometric expert dominates (58\%). In cluttered or occluded indoor environments, where neither modality is fully reliable, the fused expert contributes the highest share (47\%), enabling the model to integrate complementary cues. These findings validate the router's ability to adaptively leverage different modalities under varying 3D conditions. We will include visualizations and detailed statistics in the supplementary material.
>
> ### **(Weakness 2) Impact of 2D-to-3D Projection Error**
>   We acknowledge the reliance on camera parameters for 2D-to-3D projection. Inaccuracies caused by sensor noise or imperfect calibration can introduce projection errors, potentially leading to mislabeling background points as foreground and generating fragmented or incomplete 3D supervision signals. To address this, we apply frustum-based filtering followed by clustering to remove outliers and suppress background points. Moreover, our cross-modal MoE module adaptively balances semantic (2D) and geometric (3D) cues, enabling the model to down-weight unreliable 3D inputs and mitigate the impact of projection noise during training.
>
> In general, our method successfully localizes a wide range of objects, including those with challenging spatial arrangements such as closely placed or partially occluded instances. This is enabled by the joint use of 2D semantic priors and 3D geometric cues. Nonetheless, we observe occasional misses in cases involving objects with deformable surfaces, low color and texture contrast. This is likely caused by occasional failures of SAM and the 2D detector. We will include analyses of such cases in the revised version.

---

> > ### Comment · Reviewer_kkf7 · 2025-08-06
> > **Response to Rebuttal**
> >
> > The rebuttal addressed my concerns. I keep my rating as positive.

---

> > > ### Author Response · Authors · 2025-08-06
> > >
> > > Thank you for taking the time to review the rebuttal and the positive score. Your thoughtful input is greatly appreciated and helped us improve the quality of our work. We will incorporate your valuable feedbacks into the final version of the paper.

---

### Official Review · Reviewer_PU8d · 2025-06-30

**Clarity:** 2
**Significance:** 3
**Originality:** 3
**Rating:** 4
**Confidence:** 5

**Summary:**

The paper's main contribution lies in its ability to perform 3D object detection in an open-world setting without relying on predefined object categories or text prompts. The core idea behind OP3Det is to leverage the strengths of both 2D semantic priors, derived from pre-trained 2D foundation models, and 3D geometric priors to generate class-agnostic proposals. Specifically, the method employs the Segment Anything Model (SAM) to produce 2D segmentation masks from RGB images, which are then refined using a multi-scale point sampling strategy and a pre-trained class-agnostic 2D detector. These 2D masks are then lifted into 3D proposals using the available 3D geometric information. To further enhance the detection capabilities, OP3Det integrates complementary information from both point cloud and RGB images.

**Questions:**

Even though there are significant weaknesses I fail to overlook, I still lean towards a positive rating for this submission, as the open-world setting in 3D can be valuable to the community. For the detailed questions, please refer to the weaknesses section. I may increase or decrease my rating depends on the response from the authors.

**Ethical Concerns:**

["NO or VERY MINOR ethics concerns only"]

**Final Justification:**

The rebuttal and discussions from the authors partly address my concerns. Nevertheless, I maintain my rating for the effort and other addressed points. Please incorporate the content from your rebuttal (e.g., added ablation experiments) into the final version of the submission.

**Limitations:**

yes

**Quality:**

2

**Strengths And Weaknesses:**

Strengths:
- The proposed method achieves SOTA performance compared to existing open-vocabulary 3D detection methods.
- The multi-scale point sampling strategy and the pre-trained class-agnostic 2D detector is an interesting approach to further refine these masks, leading to more accurate 3D proposals.

Weaknesses:

Despite the strengths of this paper, I have identified several significant weaknesses:

- Inadequate Task Definition & Contextualization (Sec. 1):
The paper fails to rigorously define "Open-World Prompt-free 3D Detection" or distinguish it meaningfully from prior art. While claiming novelty, it conflates concepts like "class-agnostic detection," "open-world," and "open-vocabulary" without clarifying the unique requirements of 3D open-worldness (e.g., handling partial occlusions, viewpoint invariance in 3D space, or scale ambiguity in point clouds). Crucially, it lacks a formal comparison against well-established 2D open-world detection paradigms (e.g., Detic, OWL-ViT). The distinction boils down to "it’s 3D," ignoring whether core challenges like unknown object discovery fundamentally differ beyond the modality shift. A rigorous analysis of why 3D necessitates a wholly new task definition is absent.

- Superficial Treatment of 3D-Specific Challenges (Sec. 1 & 2):
The discussion of challenges unique to 3D open-world detection is cursory. Beyond mentioning data scarcity, it neglects critical issues like severe sparsity/occlusion in point clouds affecting object completeness, ambiguity in separating foreground objects from complex 3D backgrounds (e.g., clutter, structures), or the inherent noise and registration errors in multi-modal (LiDAR-RGB) sensor fusion. Compared to dense 2D pixels, these factors drastically impact defining "objectness" in 3D. The paper acknowledges 2D richness but doesn’t deeply analyze why transferring 2D objectness concepts to sparse 3D is fundamentally harder or how their method specifically overcomes these geometric hurdles beyond using SAM and proximity.

- Insufficient Baseline Comparisons (Sec. 4):
The baselines are limited to adapted closed-world and open-vocabulary 3D detectors. Crucially, it fails to compare against relevant SOTA methods from adjacent modalities that could serve as strong baselines or ablations. This includes: (1) SOTA 2D open-world detectors (e.g., FOUND, DetPro) applied to the input images – essential to gauge the added value of 3D; (2) Multi-view or video-based 3D detectors leveraging temporal consistency for robustness; (3) Panoramic image-based 3D understanding methods; (4) 3D self-supervised/unsupervised object discovery approaches. Omitting these leaves the reader questioning if OP3Det truly advances beyond simply applying SAM to images and projecting results.

- Failure to Explore MLLM-Based Alternatives (Sec. 2 & 4):
The paper entirely ignores the potential of Multimodal Large Language Models (MLLMs) for open-world 3D understanding, a significant oversight in 2025. It doesn't compare against or discuss: (1) Using MLLMs directly (e.g., LLaVA-3D, 3D-LLM) for prompt-free 3D object recognition and description; (2) MLLMs as generators of descriptive text or region proposals for existing 3D open-vocabulary models; (3) MLLMs for fusion instead of the proposed MoE. Given the strong zero-shot capabilities of MLLMs, their absence as a baseline or discussion point weakens the claim that OP3Det’s approach (relying heavily on SAM) is optimal or state-of-the-art for prompt-free understanding.

- Inadequate Ablation Studies (Sec. 4.5):
The ablations are superficial and fail to validate core design choices critically. Crucially, they do not ablate the necessity of the 3D (point cloud) modality itself. Removing 3D point cloud input (testing RGB-only model's performance) is essential to demonstrate if 3D geometry is truly indispensable for their defined "objectness" or if strong 2D features (SAM + MoE) suffice. Furthermore, ablations lack: (1) Testing the impact of different 2D foundation models beyond SAM; (2) Removing the MoE router to show the value of dynamic fusion vs. static fusion; (3) Quantifying the contribution of the 2D detector post-processing independently. Without these, the claimed benefits of multi-modal 3D fusion and the MoE architecture remain weakly supported.

---

> ### Author Rebuttal · Authors · 2025-07-31
>
> ### **(Weakness 1) Task Definition & Contextualization for "Open-World Prompt-free 3D Detection"**
>
>   (1) "Class-agnostic detection" refers to the task of detecting object instances without predicting their specific category labels. It simply determines whether a region contains an object. "Open-vocabulary detection" assumes access to category names or prompts and aims to recognize objects from an open set, often including unseen classes. "Open-world detection" emphasizes generalization to unknown categories. Our proposed task, open-world prompt-free 3D detection, is a class-agnostic formulation that avoids any use of semantic labels or prompts. Instead, it focuses on learning 3D objectness with the goal of detecting all object instances, especially novel ones. Grounded in 3D detection, it contributes to the broader vision of open-world learning through a semantic-free, discovery-driven perspective.
>
>   (2) Learning 3D objectness poses fundamentally different challenges compared to 2D objectness. Unlike in 2D vision, where object boundaries are often well-defined in dense pixel grids, 3D point clouds are sparse, noisy, and incomplete due to occlusions and sensor limitations, making it significantly harder to infer objectness. Furthermore, 3D datasets are scarce, manually annotated, and limited in category diversity, in contrast to the abundance of large-scale image-text datasets available for 2D. These fundamental differences make the direct transfer of 2D open-world methods like Detic or OWL-ViT to 3D neither straightforward nor effective. We will clarify these in the revised version.
>
> ### **(Weakness 2) Superficial Treatment of 3D-Specific Challenges**
>
>  While our original submission emphasized data scarcity and modality fusion, our model was in fact designed with these 3D-specific challenges, aiming to better learn 3D objectness. For the sparsity problem, we utilize a voxel-based architecture that supports learning from sparse point clouds. For occlusion and foreground-background ambiguity, we incorporate RGB images that anchor dense 2D semantics into 3D space. The multi-modal paradigm allows the model to effectively resolve incomplete geometry, missing parts, and ambiguous object boundaries, which fundamentally motivates our choice of a multi-modal framework.
>
>  We also emphasize that transferring objectness from 2D to 3D is non-trivial. Unlike dense RGB pixels, 3D data is incomplete, lacks uniform resolution, and contains structural noise, making direct application of 2D paradigms inadequate without careful cross-modal alignment and fusion.
>
> ### **(Weakness 3) Insufficient Baseline Comparisons**
>
> To further demonstrate the effectiveness of our OP3Det, we re-implement more recent methods as baselines and compare them, as shown in the table below. The 2D open-world method DetPro[1], when combined with 2D-to-3D projection, underperforms compared to our OP3Det (72.4% vs. 78.8%). This is because the projection of 2D masks into 3D introduces information loss and geometric errors, and such methods fail to exploit the complementarity of RGB and point cloud modalities. Multi-view 3D detectors such as ImVoxelNet[2] and NeRF-Det[3] also yield much lower performance (34.2% and 37.5%, respectively), as they rely solely on 2D images. These methods do not utilize 3D point clouds, where abundant 3D information exists, making them ill-suited for precise 3D object detection under open-world conditions. We also include PointContras[4], a representative 3D self-supervised approach, which achieves AR$_{novel}$ of 73.3% but is still outperformed by OP3Det. This is because PointContrast is designed for representation learning, not open-world object detection, and lacks both explicit 3D objectness modeling and localization knowledge. Together, these results demonstrate that OP3Det is not merely “SAM + projection,” but a carefully designed framework that addresses the core challenges of 3D open-world detection over multi-modal inputs.
>
>   | method                 | AR$_{novel}$ | AR$_{all}$ | AR$_{base}$|
>   |------------------------|--------------|------------|------------|
>   | DetPro + Projection (SOTA 2D open-world)   | 72.4     |  85.3  |     89.4      |
>   |ImVoxelNet (multi-view) | 34.2        |	 62.6     |	  70.8.     |
>   |NeRF-Det (multi-view) | 37.5        |	 64.2     |	  71.4     |
>   | PointContrast (3D self-supervised) |   73.3 |  82.8 | 85.7  |
>   | **OP3Det (ours)**      | **78.8**    | **89.7**   | **93.1**    |
>
> ### **(Weakness 4) Failure to Explore MLLM-Based Alternatives**
>
> (1) Existing MLLMs such as LLaVA-3D[6] and 3D-LLM[7] are not directly applicable baselines for our task. Despite recent progress, most of the existing 3D MLLMs still cannot accept raw point clouds as input. Instead, they rely on intermediaries such as multi-view images or depth images to process 3D information. Therefore, in terms of input modality, these methods are difficult to apply directly to our scenario.
>
>  (2) We argue that current MLLMs are typically not suitable tools for generating dense region proposals in our setting. These models are inherently prompt-driven and operate over 2D image inputs. Given that a user query is conditioned on language, they typically output only a small number of semantic-level bounding boxes or region captions and cannot generate dense region proposals with confidence scores required for 3D object discovery. In comparison, our method targets prompt-free, class-agnostic 3D object discovery by generating hundreds to thousands of candidate regions per scene to achieve high recall, a capability that current MLLMs are not equipped for.
>
>  (3) While MLLMs excel at vision-language tasks, they lack the ability to perform fine-grained, token-level fusion of RGB and point cloud features. Our **MoE** module is specifically designed for this purpose, enabling spatially adaptive fusion based on scene context. It dynamically adjusts modality weights, which is crucial for accurate, prompt-free 3D detection.
>
> ### **(Weakness 5) Inadequate Ablation Studies**
>
> **(1) Models beyond SAM.** While our pipeline is built on SAM due to its strong generalization ability and wide adoption, we agree that evaluating alternative or improved segmentation backbones can provide additional insight. To this end, we conducted an experiment replacing the default SAM with HQ-SAM[5], a recent enhanced variant that introduces special tokens and global-local feature fusion for improved mask quality. All other components of the pipeline were kept unchanged. These findings validate the flexibility of our pipeline and indicate that stronger 2D priors can translate into better 3D detection performance.
>
> | 2D Foundation Model | AR$_{novel}$ | AR$_{all}$ | AR$_{base}$ | AP   |
> |---------------------|:------:|:-----------:|:----------:|:----:|
> | SAM                 | 78.8                      | 89.7                     | 93.1                      | 65.4  |
> | HQ-SAM              | **79.1**                   | **90.2**                  | **93.6**                   | **65.8** |
>
> **(2) Necessity of the 3D point cloud modality.** To assess the indispensability of 3D input, we ablate the point cloud modality and only use RGB images as inputs. As shown in the Table, removing 3D leads to a significant performance drop (e.g., AR$_{novel}$ drops from 74.4% to 38.4%), especially on novel categories. This confirms that 2D alone lacks sufficient spatial and structural information to model 3D objectness in open-world 3D scenes, where distinguishing foreground from cluttered backgrounds is particularly difficult.
>
>
> | Modality | AR$_{novel}$ | AR$_{all}$ | AR$_{base}$ |
> |:---:|:-------------:|:-----------:|:------------:|
> | PC | 68.8         | 86.0        | 91.4         |
> | Img| 38.4         | 64.4        | 72.5         |
> | PC + Img | **74.4**     | **87.9**    | **92.2**     |
>
>
> **(3) Dynamic fusion (MoE) vs. static fusion.** To validate our fusion design, we compare our Cross-Modal Mixture-of-Experts (**CM-MoE**) with static strategies like addition and concatenation. As shown in the Table, CM-MoE consistently outperforms the baselines (e.g., AR$_{novel}$: 74.4% vs. 66.0%/65.4%), demonstrating its ability to dynamically adjust to modality-specific reliability.
>
>
>   | Method         | AR$_{novel}$ | AR$_{all}$ | AR$_{base}$ |
>   |----------------|:-------------:|:-----------:|:------------:|
>   | addition        | 65.4         | 85.6        | 91.4         |
>   | concatenation   | 66.0         | 85.8        | 92.1         |
>   | CM-MoE          | **74.4**     | **87.9**    | **92.2**     |
>
> **(4) 2D detector post-processing.** As shown in the Table, adding the 2D class-agnostic detector (LDET)[8] as a post-processing step to SAM + multi-scale point sampling yields clear performance gains:
>   AR$_{novel}$ improves from 61.9% to 66.1%, AR$_{all}$ from 54.2% to 59.2%, and AP from 7.6% to 10.0%.
>
>
>   | Method                          | AR$_{novel}$ | AR$_{all}$ | AR$_{base}$ | AP   |
>   |---------------------------------|:-------------:|:-----------:|:------------:|:----:|
>   | w/o 2D detector post-processing  | 61.9         | 54.2        | 51.9         | 7.6  |
>   | w/ 2D detector post-processing  | **66.1**     | **59.2**    | **57.1**     | **10.0** |
>
>
> **Reference**
>
> [1] Learning To Prompt for Open-Vocabulary Object Detection With Vision-Language Model, CVPR 2022
>
> [2] ImVoxelNet: Image to Voxels Projection for Monocular and Multi-View General-Purpose 3D Object Detection, WACV 2022
>
> [3] NeRF-Det: Learning Geometry-Aware Volumetric Representation for Multi-View 3D Object Detection, ICCV 2023
>
> [4] PointContrast: Unsupervised Pre-training for 3D Point Cloud Understanding, ECCV 2020
>
> [5] Segment Anything in High Quality, NeurIPS 2023
>
> [6] LLaVA-3D: A Simple yet Effective Pathway to Empowering LMMs with 3D-awareness, 2024
>
> [7] 3D-LLM: Injecting the 3D World into Large Language Models, NeurIPS 2023
>
> [8] Learning to Detect Every Thing in an Open World, ECCV 2022

---

> ### Author Response · Authors · 2025-08-06
>
> Dear Reviewer,
>
> Thank you for your very helpful and constructive review. We hope that our rebuttal has addressed your concerns. We are happy to answer any follow-up questions you might have. If your major concerns have been addressed properly, we would kindly appreciate it if you could consider updating your score accordingly.
>
> Sincerely,
>
> Authors.

---

> > ### Comment · Reviewer_PU8d · 2025-08-07
> >
> > I appreciate the authors' clarification, but key issues remain unaddressed:
> >
> > 3D objectness does not fundamentally different with its 2D conterpart, i.e., class-agnostic open-world 2D detection. The class-agnostic (prompt-free) task is not novel: Prior 2D class-agnostic detectors, like [55], already operate without category prompts. Moreover, according to the additional experiment results for (Weakness 3) Insufficient Baseline Comparisons, the existing 2D open-world method DetPro + Projection, as well as 3D self-supervised method PointContrast can produce comparable performance. They are even without training specifically on 3D indoor scene and autonomous driving datasets as used in this paper, SUN RGB-D, ScanNet V2 and KITTI, and this speaks for their slightly worse performance. The core innovation should be how 3D geometry redefines objectness itself, yet the rebuttal only cites data limitations. I suggest discussing more on fundamental geometric or topological properties unique to 3D.

---

> > > ### Author Response · Authors · 2025-08-08
> > > **Thanks for your comments! Any new questions please let me know!**
> > >
> > > In our work, 3D objectness is defined as the likelihood that a spatial region corresponds to a physically discrete object in 3D space, distinguishable from background structures or noise, regardless of its semantic category. This is modeled by a learnable function $\phi(\cdot)$ that maps input features $F_{input}$ to an objectness score, and makes the foreground/background decision via thresholding:$I \left[ \phi(F_{input}) > \tau \right]$, where $\tau$ is the threshold for foreground-background separation; $I[\cdot]$ is the indicator function (1 for foreground, 0 for background).
> > >
> > > Concretely, our model predicts a set of 3D proposals $B_{3D} = $\{$b_{3D}|b_{3D} = (x, y, z, w, l, h, θ)$\}, where $(x, y, z)$ is the center and $(w, l, h, θ)$ are the width, length, height, and yaw angle respectively. For each 3D proposal, the model estimates an objectness score to determine whether it corresponds to a true object or background clutter. In contrast, 2D bounding boxes $B_{2D}=$ \{$ b_{2D}|b_{2D} = (x,y,w,l)$\}, directly projecting them into 3D space inevitably introduces distortion. While image features convey rich texture and appearance cues, they inherently lack explicit geometric structure. By comparison, voxel features provide a spatially regular representation of point cloud data, where each voxel corresponds to a grid-size 3D space. This representation naturally preserves accurate localization and detailed geometric information, motivating our approach to exploit the complementary strengths of both image and point cloud modalities.
> > >
> > > However, the distinct data distributions of image and point cloud modalities pose challenges for effective fusion. Mapping both into a shared representation simplifies integration but often leads to information loss, such as missing height cues in the image or occlusions and distortions in the point cloud. Object-level fusion captures high-level semantics but heavily depends on feature quality, potentially compromising robustness and objectness learning. To address this, we introduce a cross-modal Mixture of Experts (MoE) module that adaptively integrates both intra- and inter-modal features. This allows the model to selectively attend to the most informative cues across modalities, ensuring more accurate and reliable object representation.
> > >
> > > We conducted a quantitative analysis of the expert weight distribution across diverse scenes and object types. Results demonstrate clear and meaningful patterns: the semantic expert is favored in high-texture scenes (averaging 62% of the weight), the geometric expert is prioritized in sparse or textureless regions (58%), and the fused expert dominates in cluttered indoor environments (47%). These trends are consistent with intuition and confirm that the router does not apply a static weighting scheme, but rather dynamically adjusts based on the local reliability of each modality for different scenes. With these designs, the learned model $\phi(\cdot)$ can be more robust, making $I \left[ \phi(F_{input}) > \tau \right]$ more accurate. We will incorporate your valuable feedbacks into the final version of the paper.

---

### Official Review · Reviewer_99tt · 2025-07-06

**Clarity:** 2
**Significance:** 3
**Originality:** 2
**Rating:** 4
**Confidence:** 3

**Summary:**

This paper proposes a class-agnostic open-world prompt-free 3D Detector for 3D objectness learning. The is framework adopts RGB image and point cloud as input, then outputs class-agnostic 3D bounding boxes. In the training pipieline, a 3D object discovery mechanism is used to cover more object instances in the training data. Then a cross-modal mixture of experts fusion module is proposed to fuse features from image and point cloud. Finally, a 3d object detector Uni3DETR is used to predict final outputs. The authors conduct experiments on SUN RGB-D and ScanNet dataset and compare the proposed method with other closed-world or open-vocabulary 3D object detection methods. The experiments shows that the proposed class-agnostic detector can achieve higher performances comparing to other methods.

**Questions:**

1. Please add more evaluation details in the rebuttal.
2. Why did the authors need to design a new detector instead of using the expanded 3D bounding boxes to train an existing closed-world or open-vocabulary detector?

**Ethical Concerns:**

["NO or VERY MINOR ethics concerns only"]

**Final Justification:**

The rebuttal address my concerns. I keep my rating as positive.

**Limitations:**

Yes

**Paper Formatting Concerns:**

No formatting issues.

**Quality:**

3

**Strengths And Weaknesses:**

Strengths:
1. This paper focus on a interesting task, 3D objectness learning and proposed a pipeline which can obtain high performances class-agnostic 3d bounding boxes with multi-modal inputs.
2. The experiments show the proposed method achieve a large margin performance improvements comparing to other closed-world or open-vocabulary 3D object detection methods.

Weaknesses:
1. Missing evaluation details.
- How many objects are retained for evaluation when computing the Average Recall (AR) metric?
- Is the lower AR in closed-world/open-vocabulary detectors due to false negatives (FN) being filtered out by a high confidence threshold?
2. Necessity of a new detector
- Could the proposed 3D object discovery pipeline be used to expand training data, followed by fine-tuning an existing closed-world/open-vocabulary detector (instead of designing a new one)?
- Would integrating SAM3D [46] with the proposed point-sampling strategy yield comparable or better 3D bounding boxes?

---

> ### Author Rebuttal · Authors · 2025-07-30
>
> ### **(Weakness 1.1) Quantifying object candidates in AR computation**
>
> We retain **1,000 predicted objects per scene** when computing AR, following common practice in 3D detection benchmarks. This ensures a fair comparison across methods. We will clarify such evaluation details in the final version.
>
>
> ### **(Weakness 1.2) On the role of confidence thresholds in AR evaluation**
>
> For evaluation, we follow prior open-vocabulary 3D detection works (e.g., OV-Uni3DETR) and adopt a standardized post-processing strategy where a **fixed number of top-K predictions (1000 per scene)** and the **same confidence threshold (0.01)** are retained for AR computation.
>
> Therefore, the lower AR observed in closed-world or open-vocabulary detectors is not due to high thresholds filtering out correct predictions, but rather reflects the actual limitations of these models in generating high-recall proposals for novel categories. We will clarify this in the final version.
>
>
> ### **(Weakness 2.1 and Question 2) Adaptability of 3D object discovery & Necessity of a new detector**
>
> Yes, our 3D object discovery pipeline can be leveraged to expand training data, especially for previously unseen object categories. With a broader range of annotated objects, the discovered 3D regions can be used to fine-tune existing closed-world or open-vocabulary detectors.
>
> To demonstrate this, we conducted experiments where we used our discovered objects to augment training data and fine-tune existing detectors such as FCAF3D[1] and Tr3D[2]. As shown in the table below, incorporating discovered objects leads to substantial improvements in AR$_{\text{novel}}$:
>
> | Method                        | AR$_{\text{novel}}$ | AR$_{\text{all}}$ |
> |------------------------------|--------------------|------------------|
> | FCAF3D                       | 65.3               | 86.5             |
> | FCAF3D + object discovery    | 74.9               | 88.4             |
> | Tr3D                         | 62.1               | 84.8             |
> | Tr3D + object discovery      | 72.9               | 85.1             |
> | **OP3Det (ours)**            | **78.8**           | **89.7**         |
>
> Although existing detectors benefit from our discovery pipeline\, **OP3Det further outperforms them**. It is worth noting that both FCAF3D and TR3D are multi-modal models, showing that a newly designed detector is necessary for effective multi-modal fusion in an open world.
>
>
> ### **(Weakness 2.2) Evaluating the compatibility of our point-sampling strategy**
>
>   We agree that combining SAM3D[3] with our proposed point-sampling strategy could be a promising direction. To validate this, we conducted an ablation study where we integrated the SAM3D strategy with our multi-scale point sampling method before 3D box generation.
>
> | Method                            | AR$_{novel}$ |
> |----------------------------------|-------------------------|
> | SAM3D                             | 58.6                    |
> | SAM3D + multi-scale sampling      | 61.7                    |
> | **Ours (SAM + multi-scale sampling)** | **62.8**                |
>
> As shown in the table above: SAM3D alone performs worse due to unfiltered noisy masks containing clutter; Adding our point-sampling strategy improves SAM3D performance (from 58.6% to 61.7%); Our full pipeline still outperforms (62.8%) due to additional refinement from the 2D class-agnostic detector, which helps filter non-object regions before 3D lifting.
>
> These findings support the generalizability and effectiveness of our point-sampling method for robust 3D box generation.
>
> **Reference**
>
> [1] FCAF3D: Fully Convolutional Anchor-Free 3D Object Detection, ECCV 2022
>
> [2] TR3D: Towards Real-Time Indoor 3D Object Detection, ICIP 2023
>
> [3] SAM3D: Segment Anything in 3D Scenes, ICCV 2023 Workshop

---

### Note · Authors · 2025-08-16

We sincerely thank all reviewers for their valuable and constructive feedback. We are encouraged by your positive assessment of our contributions and greatly appreciate the comments highlighting the strengths of our work.

- Propose a novel and valuable task — learning 3D objectness in a class-agnostic manner, aiming to detect all objects in diverse real-world 3D scenarios.
- Introduce a multi-scale point sampling strategy with a pre-trained class-agnostic 2D detector as a training-free approach for 3D object discovery. Ablation studies demonstrate the necessity and effectiveness of these components, as well as the benefits of dynamic multi-modal fusion.
- Conduct comprehensive experiments covering cross-category settings, cross-dataset settings, and both indoor and outdoor scenarios, showing significant improvements over existing methods in challenging generalization settings.

We have carefully addressed each individual comment provided by the reviewers and believe we have successfully responded to most of their concerns. In the revised version, we will:

- Provide a mathematical formulation of 3D objectness that naturally reflects its fundamental geometric properties, along with a discussion of 3D-related challenges beyond data limitations and the clarification of the "prompt-free" setting.
- Add analysis of the MoE module's adaptive behavior and the detailed comparisons in the outdoor cross-dataset setting (KITTI → nuScenes).
- Include experiments with SAM3D + multi-scale sampling and HQ-SAM + multi-scale sampling, where the 2D foundation model is replaced with alternatives beyond SAM, demonstrating both the compatibility of the multi-scale sampling strategy and its standalone contribution.
- Report the number of object candidates in AR computation and analyze computational cost scaling.

We believe these additions and clarifications comprehensively address the reviewers' concerns and enhance the overall quality of the manuscript. We will further refine and expand the supplementary material to present our methods and results more clearly and comprehensively.

---

### Decision · Program_Chairs · 2025-09-17

**Decision:**

Accept (poster)

**Comment:**

The paper receives four acceptance rating. Initially, the reviewers have concerns about some technical clarity (e.g., task definition) and experimental results (e.g., baseline comparisons, ablation study, computational analysis). After the rebuttal, despite the concerns about the 3D objectness definition, all the reviewers are satisfied with the addressed issues in the rebuttal. The AC closely checks the paper, reviews, rebuttal, and the discussed points, and agrees with the assessment from reviewers. Therefore, the AC recommends the acceptance rating and encourages the authors to incorporate the suggestions raised by the reviewers in the final version, as well as releasing the model and code for reproducibility.